# Riverine nutrient impact on global ocean nitrogen cycle feedbacks and marine primary production in an Earth System Model

Miriam Tivig[1,2], David P. Keller[1,3], and Andreas Oschlies[1]

[1]GEOMAR Helmholtz-Zentrum für Ozeanforschung Kiel, Wischhofstr. 1-3, D-24148 Kiel, Germany
[2]Deutscher Wetterdienst, Abteilung Klima und Umwelt, Michendorfer Chaussee 23, D-14473 Potsdam, Germany.
[3]Carbon to Sea Initiative

**Correspondence:** Miriam Tivig (mtivig@geomar.de)

**Abstract.**

Riverine nutrient export is an important process in marine coastal biogeochemistry and also impacts global marine biology. The nitrogen cycle is a key player here. Internal feedbacks are shown to regulate not only nitrogen distribution, but also primary production, and thereby oxygen concentrations. Phosphorus is another essential nutrient and interacts with the nitrogen cycle

via different feedback mechanisms. After a previous study of the marine nitrogen cycle response to riverine nitrogen supply, here we include phosphorus from river export with different phosphorus burial scenarios and study the impact of phosphorus alone and in combination with nitrogen in a global 3-D ocean biogeochemistry model. Again, we analyse the effects on near-coastal and open-ocean biogeochemistry. We find that riverine export of bioavailable phosphorus alone or in conjunction with nitrogen affects marine biology on millennial timescales more than riverine nitrogen alone. Biogeochemical feedbacks in the

marine nitrogen cycle are strongly influenced by additional phosphorus. Where bioavailable phosphorus increases with river input, nitrogen concentration increases as well, except for regions with diminishing oxygen concentrations. High phosphorus burial rates decrease biological production significantly. Globally, the addition of riverine phosphorus in the modelled ocean leads to elevated primary production rates in the coastal and open oceans.

**Plain Language Summary**

Coastal oceans are the most productive parts of the global ocean and, arguably, most sensitive to environmental change, but poorly resolved in most global models. In reality, rivers influence coastal oceans because they transport nutrients from land to sea. This nutrient supply is often not included in global models, even though it may impact not only the coastal biology, but also the marine biology in regions farther away from the river. We here include dissolved inorganic nitrogen and phosphorus from river export in a global ocean model and study the effects on the near coastal and the open ocean. We find that the addition

of riverine phosphorus affects marine biology on millennial timescales more than riverine nitrogen alone. Where phosphorus increases with river input, nitrogen concentrations increase, as well, except for regions with high rates of bacterial nitrate consumption (denitrification). High phosphorus burial rates on the other hand, decrease biological production significantly. Globally, riverine phosphorus leads to higher primary production rates in the coastal and open oceans.

## 1 Introduction

Nitrogen and phosphorus are both considered as limiting nutrients in the global ocean. 'Limiting' refers here to the concept of Liebig's law of the minimum, i.e. the growth rate beeing determined by the availability of the most limiting substrate (Allaby, 2010). It states that growth is dictated not by total resources available, but by the scarcest resource (limiting factor). Nitrogen can be considered as the 'proximate limiting' and phosphorus as the 'ultimate limiting' nutrient, according to the definition by

Tyrrell (1999). Changes in the availability of oceanic fixed-nitrogen (N) are known to have driven marine productivity changes, thereby regulating the strength of the biological carbon pump. By this, the availability of N also influences the carbon cycle in the atmosphere and in the ocean (Falkowski, 1997). Phosphorus (P) limits marine productivity on geological and global scales and plays an important role in regulating oceanic oxygen inventories (Monteiro et al., 2012; Palastanga et al., 2011).

Oceanic fixed nitrogen concentrations are mainly controlled by the balance between nitrogen fixation and denitrification, but

atmospheric deposition and riverine input also contribute to the global N budget (Somes et al., 2013; Deutsch et al., 2007; Gruber, 2004; Ruttenberg, 2003). Although several studies question the stability of the marine N inventory (Zehr and Capone, 2020; Codispoti et al., 2001; Gruber and Sarmiento, 1997; Codispoti, 1995), the pre-industrial global nitrogen cycle is often assumed to reflect a steady state (Deutsch et al., 2007; Altabet, 2006; Gruber, 2004; Tyrrell, 1999; Redfield et al., 1963).

This balance is assured by negative N-cycle feedbacks that stabilize the marine N inventory: Where fixed N is sparse, dia-

zotrophs can fix atmospheric $N_2$ instead of depending on dissolved inorganic nitrogen (DIN), such as ammonia or nitrate. But as the process of atmospheric N fixation is slow and requires more energy, mostly to keep oxygen away from the oxygen-sensitive enzyme nitrogenase, diazotrophs are rapidly outcompeted by other phytoplankton who do not have this energetic cost, if sufficient DIN is available. Growth of diazotrophs is also limited by the availability of phosphate, light and iron.

On the other hand, the global marine N budget is regulated by loss of fixed N, predominantly via denitrification. This process

describes anaerobic respiration of organic matter via bacterial reduction of nitrate to $N_2$, and occurs in the water column as well as in sediments if the oxygen concentration is low (Gruber, 2004; Deutsch et al., 2001). Denitrification limits itself, as the consumption of nitrate leads to a reduction in the production of organic matter and hence to less oxygen consumption.

Generally the processes of $N_2$-fixation and denitrification take place in different regions of the world ocean. Still, feedbacks link these processes globally and are generally assumed to restore the balance in the global marine N budget. In some regions,

however, $N_2$-fixation and denitrification occur in geographical proximity and may produce a "vicious cycle" with a local run-away loss of fixed N (Landolfi et al., 2013). Estimates of the mean residence time of fixed nitrogen in the global ocean amount to a few thousand years (Gruber, 2004). Due to the complexity of internal feedbacks and the dynamic role of nitrogen in the marine biological production it is difficult to assess, how sensitive the global N concentration is to perturbations of the marine biogeochemistry.

In a previous study (Tivig et al., 2021), we used the Earth system climate model of intermediate complexity of the University of Victoria (UVic) version 2.9 (Eby et al., 2009; Weaver et al., 2001) to study N cycle feedbacks in the modelled ocean in

response to the addition of riverine dissolved inorganic nitrogen. We found that, although in hot-spots near the river mouths marine primary production increased due to the additional N, globally biogeochemical feedbacks stabilized N concentrations and primary production, and could even lead to a local decline in N and productivity in proximity to low oxygen regions. In

those idealized simulations, N was the only nutrient supplied via rivers.

With this, we confirmed an early study by Tyrrell (1999), who used a box model and found, that "an increase in the river delivery of nitrate has no long-term effect on productivity". But Tyrrell (1999) also concluded, that "an increase in the river delivery of phosphate, on the other hand, causes a sustained and proportionate increase in productivity". The interaction between fixed N and P is of particular interest for marine primary production. The availability of P is one of the limiting factors

for $N_2$-fixation (Wang et al., 2019; Landolfi et al., 2015).

In Tivig et al. (2021) we also found, that locally, marine productivity was increased by riverine N supply. Riverine supply of P is predicted to regulate total ocean productivity globally (Tyrrell, 1999). Nevertheless, locally, in coastal oceans and in regions with low oxygen concentrations, the sensitivity of marine biogeochemistry to perturbations of the nutrient cycle can differ from global averages.

The total P inventory in the global ocean is mainly controlled by riverine input and burial at the seafloor (Wallmann, 2010; Ruttenberg, 2003; Baturin, 2003; Delaney, 1998). The residence time of P in the global ocean is approximately one order of magnitude longer than that of fixed N (Delaney, 1998). (Bio-)chemical weathering on continents represents the main source of riverine bio-available P (Filipelli, 2008; Föllmi, 1995). Changes in the marine P inventory are mostly a consequence of changes in terrestrial weathering and therefore have generally occurred on geological timescales.

Nevertheless, climate change and direct anthropogenic interventions in the P cycle have started to alter P fluxes during the last century. Due to the wide-spread use of fertilizers, to deforestation, and sewage sources, the riverine load of phosphorus has increased globally (Seitzinger et al., 2010; Filipelli, 2008). In addition, river flows have also been dramatically altered by land use change and the damming of rivers (Cappellen and Maavara, 2016). In order to predict future changes in marine biogeochemistry, it is therefore relevant to understand, how P fluxes impact the marine N cycle, the N cycle feedbacks and

marine productivity globally and regionally.

As global observations and measurements of ocean nutrients and fluxes are difficult and observations still relatively sparse, models are often used to investigate large-scale marine biogeochemistry over long time scales. However, only recently have there been more global modelling studies with riverine nutrient input. In one of the first studies, Giraud et al. (2008) analysed coastal fluxes of P, silicate, and dissolved iron in a global ocean model and found that including nutrients in the coastal ocean

impacts biological activity locally but also in the open ocean. They also found that excess nutrients in the coastal ocean can impact the open ocean biogeochemistry depending on which nutrient is advected from the coastal region. Nutrient availability and its consumption in the coastal domain control this transport. Additional P does affect coastal oceans especially if they are P-limited and if they are not limited by other nutrients like iron. In this case, Giraud et al. (2008) found that increased primary production in the coastal oceans can lead to a depletion of nutrients in the open ocean, reducing biological activity there. This

"seesaw effect" was detected by Giraud et al. (2008) on local and global scale. If P is not consumed in the coastal oceans, it may be advected offshore, eventually increasing primary production there. N was not simulated explicitly in that study but

coupled to P via the Redfield ratio. The simulation performed by Giraud et al. (2008) was only run for 10 years, hence not long enough for N-cycle feedbacks to fully materialise.

In a more recent study (Lacroix et al., 2020) implemented estimated riverine nutrient loads of P, N, iron, carbon (C) and silica
in a global ocean model and compared the results with those of a reference simulation, where the same nutrients were added directly and homogeneously to the open ocean surface. They found that even if the ocean circulation remains the main driver for biogeochemical distributions in the open ocean, it appeared necessary to include riverine inputs for the representation of heterogeneous features in the coastal ocean. They identified the catchments of the tropical Atlantic, the Arctic Ocean, South-east Asia and Indo-Pacific islands as regions of dominant contributions of riverine supplies to the ocean, leading to a strong
primary production increase in the tropical west Atlantic, the Bay of Bengal and the East China Sea. Nevertheless the focus in this study was mainly on C export, and N feedbacks were not considered by Lacroix et al. (2020).

As P concentrations have been increasing in many water bodies over the world, rivers transport more P to the coastal oceans. Beusen and Bouwman (2022) showed that human-dominated river supply of N and P has not only increased in the past, but will do so in the future, due to legacies of past nutrient management, even if efforts are made to reduce these nutrient loads.
Regarding P, not only is the addition of nutrient *per se* relevant for ocean marine biogeochemistry, but the stoichiometric ratio of N and P is also essential (Garnier et al., 2010; Redfield et al., 1963; Beusen and Bouwman, 2022). In the present study we have extended our experiment from Tivig et al. (2021) and include riverine supply of P in addition to N. We again aim to study the feedbacks in the N-cycle but this time face to the combined input of both limiting nutrients.

Specifically, we address the following questions:

– How does riverine N and P input together, rather than riverine N input alone, affect the representation of ocean biogeo-chemistry including marine primary production in our model?

– How does the addition of riverine P input affect specifically the N cycle and N-cycle feedbacks?

– What effect has the inclusion of riverine P fluxes specifically on marine oxygen concentrations?

To address these questions, an Earth system model of intermediate complexity (Claussen et al., 2002) that resolves the relevant
biogeochemical feedbacks is employed. It allows the integration of a large number of processes at reduced computational costs due to a coarser resolution and simplified assumptions, in case of UVic with respect to atmospheric dynamics. This type of model makes it possible to run simulations on millennial scales with different assumptions, allowing the analysis of processes and feedbacks operating in the climate system on such timescales (Weaver et al., 2001). Compared to the study of Tyrrell (1999), where a simple box model was used to study the relative influences of nitrogen and phosphorus on oceanic primary
production, the current study is based on a global 3D Earth system model, where the global and regional distribution of N and P can be analysed in more detail. This is the prerequisite for the inclusion of river exports to the ocean, with different amounts of nutrients depending on the individual rivers. Hence, regional feedbacks in the N cycle and their effects on marine primary production can be assessed as well as localised biogeochemical responses for example near oxygen minimum zones.

## 2 Methods

### 2.1 Earth system model UVic

The University of Victoria Earth System Climate model (UVic ESCM) version 2.9 (Eby et al., 2009; Weaver et al., 2001) consists of a three-dimensional (1.8° x 3.6°, 19 levels) general circulation model of the ocean, a two-dimensional, single-layer energy-moisture balance atmospheric model, a dynamic-thermodynamic sea ice model, and a terrestrial vegetation model. The atmospheric energy–moisture balance model (Fanning and Weaver, 1996) dynamically calculates heat and water fluxes between the atmosphere and the ocean, land and sea ice, and is forced by monthly climatological winds prescribed from NCEP/NCAR. The 19 vertical levels of the oceanic component, Modular Ocean Model 2 (MOM2), are 50 m thick near the surface and up to 500 m in the deep ocean. The oceanic physical settings are the same as in Keller et al. (2012). The ocean model includes a marine ecosystem module based on Keller et al. (2012) with updates as noted in Partanen et al. (2016). The ocean ecosystem and biogeochemical model is an improved NPZD (nutrient, phytoplankton, zooplankton, detritus) ecosystem model based on Schmittner et al. (2008) and includes seven prognostic variables: two phytoplankton classes (nitrogen fixing diazotrophs $P_D$ and other phytoplankton $P_O$), zooplankton (Z), sinking particulate detritus (D), nitrate ($NO_3$), phosphate ($PO_4$) and oxygen ($O_2$) (Fig. 1). $NO_3$ and $PO_4$ are linked through exchanges with the biological variables by constant (Redfield) stoichiometry (Schmittner et al., 2008). Since diazotrophs can fix nitrogen gas dissolved in seawater, they are not limited by $NO_3$, while the growth of other phytoplankton is limited by $NO_3$ and $PO_4$. All phytoplankton are additionally limited by iron, light and temperature. For the current study of nitrogen cycle feedbacks, it is a clear advantage that UVic explicitly calculates diazotrophs and $N_2$-fixation. Keller et al. (2012) found that patterns and global amounts of modelled $N_2$-fixation were mostly consistent with the relatively sparse available observations (Sohm et al., 2011). The main differences are discussed within the framework of our results. See Keller et al. (2012) for a full description and evaluation of simulated marine biogeochemistry.

As in our previous study (Tivig et al., 2021) we use empirical transfer functions derived from benthic flux measurements to calculate benthic denitrification following Bohlen et al. (2012), combined with a subgrid bathymetry scheme for shallow continental shelves and other topographical features that are too fine to be resolved on the coarse UVic grid (see details in Somes et al. (2010b)).

### 2.2 Including riverine Nitrogen and Phosphorus

#### 2.2.1 Global Nutrient Export from WaterSheds 2 (NEWS2)

The basic UVic model and ecosystem module do not account for riverine nutrient input. The only source of N in the ocean model consists in $N_2$-fixation. In Tivig et al. (2021) we included riverine N as calculated by a global, spatially explicit model of nutrient exports by rivers, NEWS2 (Mayorga et al., 2010). This second version of a system of submodels estimates the present-day annual export yield at the river mouth for each of 6081 river catchment areas and for dissolved and particulate forms of organic and inorganic N and P, as well as dissolved organic and particulate carbon. In our model study, the parameterisation of riverine N flows is identical to Tivig et al. (2021). Since nitrate is the only nitrogen nutrient explicitly resolved in the UVic

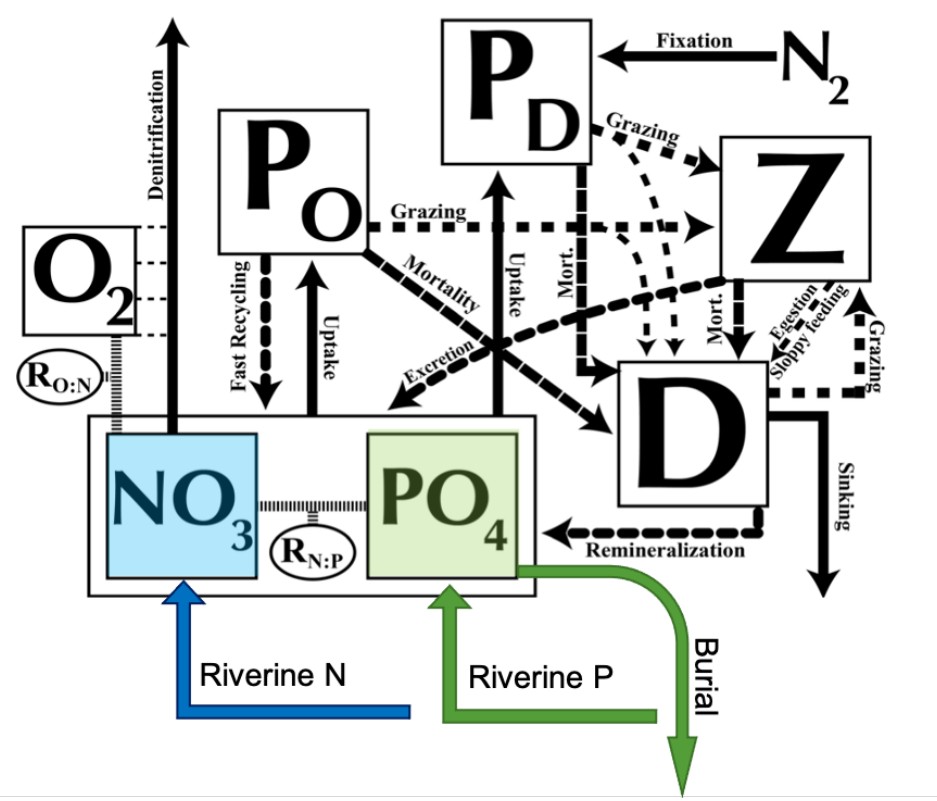

**Figure 1.** Ecosystem model schematics for the NPZD (nutrient, phytoplankton, zooplankton, detritus) model with the prognostic variables (in square boxes) and the fluxes of matter between them, indicated by arrows. See details in the text. Figure updated from Keller et al. (2012) and Tivig et al. (2021).

version used, all bioavailable N has been included in the nitrate compartment of the model. See Mayorga et al. (2010) for more details on the model configuration and Dumont et al. (2005) for more details on the validation of NEWS dissolved inorganic nitrogen (DIN).

160

### 2.2.2 Riverine reactive Phosphorus

Earlier applications of the UVic ESCM assumed a fixed marine P inventory (Keller et al., 2012; Oschlies et al., 2019; Schmittner et al., 2005). In addition to DIN, we here include also P from river discharge. We focus on the total amount of reactive P, i.e. that P that exchanges with the dissolved oceanic P-reservoir and thus is available for biological uptake (Filipelli, 2008; Ruttenberg, 2003). Estimates of globally integrated pre-industrial riverine supply of bioavailable P range from 0.1 Tmol P yr$^{-1}$ to 0.3 Tmol P yr$^{-1}$ (Kemena et al., 2019; Ruttenberg, 2003; Filipelli, 2008; Compton et al., 2000; Colman and Holland, 2000). Taking into account only dissolved inorganic P from rivers would underestimate the amount of bioavailable P from river discharge, as most

studies estimate DIP export from rivers to be significantly lower, between 0.01 and 0.05 Tmol P yr$^{-1}$ (Mayorga et al., 2010; Filipelli, 2008; Harrison et al., 2005). Following the results of Colman and Holland (2000) and Ruttenberg (2003), we decided to include dissolved organic and inorganic P (DOP, DIP), as well as 45 % of total particulate P (TPP). This represents the upper range of the fraction of the riverine TPP flux estimated as reactive P (Ruttenberg, 2003; Colman and Holland, 2000). The numbers for DIP, DOP and PP export at the river mouths have been taken from the NEWS2 data set. DIP and DOP were taken as such, PP was multiplied by 0.45 to obtain the desired fraction of total particulate P. Consistent with Tivig et al. (2021), the nutrients from NEWS2 were interpolated onto the coarser UVic grid. We assumed a periodic seasonal cycle in runoff and that concentrations in the discharged river water are constant throughout the seasonal cycle. The annual P load is thus distributed over the months using the fractions of monthly freshwater discharge as respective weights. The global amount of P we added to the UVic ocean is 0.17 Tmol P yr$^{-1}$ (5.4 Tg P yr$^{-1}$), which lays in the range estimated by previous studies (Kemena et al., 2019; Ruttenberg, 2003; Benitez-Nelson, 2000). Since phosphate is the only phosphorus nutrient explicitly resolved in the UVic version used, we decided, like for N, to put all bioavailable P into the phosphate compartment of the model.

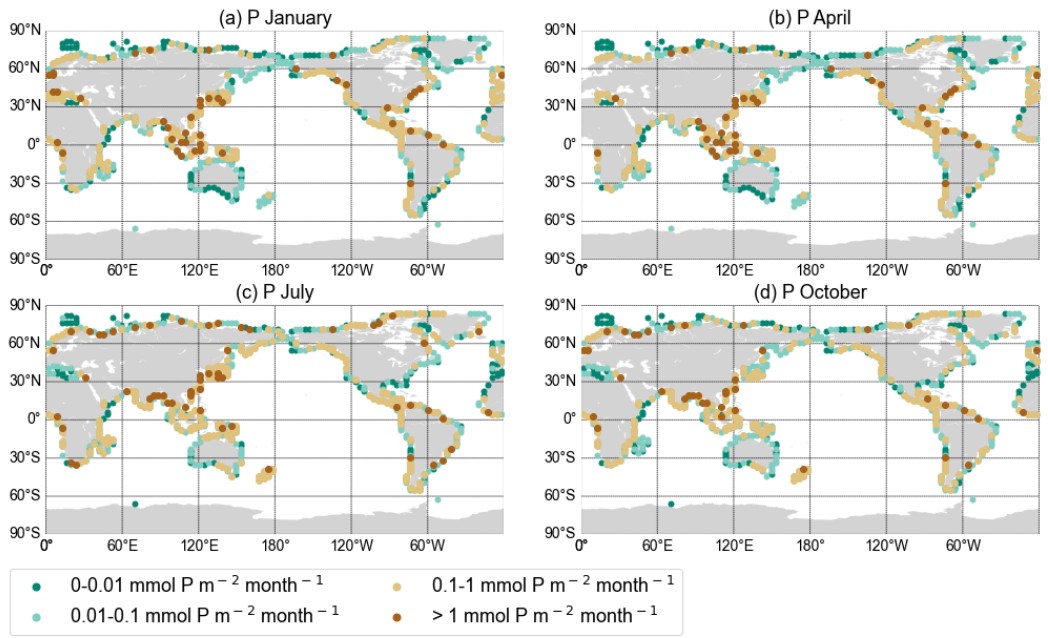

**Figure 2.** Export yield of total reactive phosphorus for each discharge point in mmol P m$^{-2}$ month$^{-1}$ from NEWS2 data set interpolated on the UVic grid for January (a), April (b), July (c) and October (d).

### 2.2.3 Burial of Nitrogen and Phosphorus

In the global ocean, N input via N$_2$-fixation and rivers is balanced by denitrification (here implicitly including anammox). In particular, denitrification in the benthic zone is considered the main sink for fixed N (Voss et al., 2013; Galloway et al., 2004).

As in Tivig et al. (2021), we include empirical transfer functions derived from benthic flux measurements (Bohlen et al., 2012) to simulate benthic denitrification. These functions are based on dynamic vertically integrated sediment models and estimate

denitrification from the rain rate of particulate organic carbon to the seafloor, bottom water $O_2$ and $NO_3$ concentrations. We use a subgrid bathymetry scheme for shallow continental shelves to better resolve particulate organic matter sinking and remineralization at the seafloor (Somes and Oschlies, 2015; Somes et al., 2013).

UVic does not contain a prognostic and vertically resolved sediment model. Therefore, the input of reactive P to the ocean has to be counterbalanced by a parameterized sink. For this purpose, we include burial functions based on Flögel et al. (2011) and

190 Wallmann (2010), which have been tested in previous studies with UVic by Kemena et al. (2019) and Niemeyer et al. (2017). With these functions, the burial of P in the sediment ($\text{BUR}_p$) is calculated in every ocean grid box column from the difference between the simulated detritus P rain rate to the sediment ($\text{RR}_p$) and the benthic release flux of phosphate from the sediment ($\text{BEN}_p$):

$$\text{BUR}_p = RR_p - BEN_p, \tag{1}$$

Following Flögel et al. (2011), the burial of P is evaluated separately for each grid box column on the shelf and in the deep-sea. Deep-sea is defined as ocean grid box columns where the ocean is deeper than 1000 m.

Benthic P release ($\text{BEN}_p$) is calculated locally as:

$$\text{BEN}_p = \frac{BEN_c}{r_{c/p}}, \tag{2}$$

$\text{BEN}_c$ represents the benthic fluxes of carbon and is computed from the difference of the carbon rain-rate to the sediment ($\text{RR}_c$)

and a virtual burial flux of organic carbon ($\text{BUR}_c$):

$$\text{BEN}_c = RR_c - BUR_c, \tag{3}$$

Depending on the ocean depth of the considered grid box, $\text{BUR}_c$ is computed from the modelled detritus export in terms of carbon, on the shelf and continental margin (Kemena et al., 2019):

$$\text{BUR}_c = 0.14 RR_c^{1.11}, \tag{4}$$

and in the deep sea:

$$\text{BUR}_c = 0.14 RR_c^{1.05}, \tag{5}$$

$\text{RR}_c$ is calculated by the model in mmol C m$^{-2}$ yr$^{-1}$. The $r_{c/p}$ ratio depends on bottom water oxygen concentration and is calculated following Kemena et al. (2019) and Wallmann (2010):

$$\text{r}_{c/p} = Y_F - A * exp(-O_2/r) \tag{6}$$

with $O_2$ in mmol m$^{-3}$ and the coefficients $Y_F = 123 \pm 24$, $A = -112 \pm 24$ and $r = 32 \pm 19$ mmol m$^{-3}$. Variations in the coefficients determine the strength of the burial. Because there are large uncertainties in these numbers, different experiments have been performed to evaluate the model response to variations in burial.

## 2.3 Experimental design

To analyse the effect of riverine nutrient supply in the UVic model, six experiments were performed (Table 1). In each experiment the model was run for 10,000 years, starting from an already-spun-up steady state with the standard model version without riverine nutrients, with preindustrial conditions for insolation and a fixed atmospheric $CO_2$ concentration of 283 ppm (Keller et al., 2012), i.e. a stable climate. Note that the spin-up did not include benthic denitrification and subgrid bathymetry. These two features have been included at the start of each simulation, including the Control Simulation (CTR). Hence, at the start of each simulation and for approximately 2000 years, there is a drift in the fixed N inventory.

Like in our previous study Tivig et al. (2021), a control simulation (CTR) was performed without riverine nutrients. The simulation NEWS-N is identical to the main experiment analysed in Tivig et al. (2021), where only DIN from river discharge was added to the coastal ocean. In a follow-up experiment, rivers exported only P to the modelled ocean (NEWS-P), but without any burial to balance the P budget. NEWS-N+P tested the impacts on marine biology and biochemistry with regard to additional N and P (without P burial). Finally, two P burial variation experiments have been performed, where riverine N and P supply from the NEWS2 model was applied. In a low burial configuration (N+P-BURLOW) we used the coefficients tested by Kemena et al. (2019) in their simulation low-bur. The second burial experiment (N+P-BURHIGH) includes the original burial functions of Flögel et al. (2011) with the coefficients for $Y_F$, $A$ and $r$ described there as well as in the reference burial experiment in Kemena et al. (2019) (Table 1).

Note that previous studies using these burial functions balanced the modelled P budget by an equivalent weathering flux providing P to the ocean via river discharge in the same amount as P was buried in the marine sediments (Kemena et al., 2019; Niemeyer et al., 2017). In the current study there is no direct link between the two fluxes: riverine P is calculated from the NEWS2 model (Mayorga et al., 2010), while P burial is calculated independently, using only detritus export and bottom-water oxygen concentration (see Table 1 for the overview of the fluxes). In all simulations where riverine P is included, the global P budget is therefore not exactly balanced (Figure 3b). While P concentrations only slowly increase in the scenario with a low burial rate, the increase is stronger in the two scenarios without P burial (NEWS-P and NEWS-N+P). In the scenario with high burial rates, P concentrations decrease and the total P flux is negative. In the timeseries (Figure 3c) it is also visible, that this flux is slowly increasing over the simulation and hence the P inventory is following an asymptotic evolution.

These different simulations permit us to analyse the sensitivity of the N cycle feedbacks under different conditions of P supply and do not pretend to reproduce the exact reality. So far, the phosphorus balance of the (pre-anthropogenic) ocean is poorly defined, and the input and output fluxes are only rudimentarily constrained (Wallmann, 2010; Föllmi, 1995).

## 3 Results

### 3.1 Phosphorus dynamics

In the four simulations with riverine P input, globally 0.17 Tmol P yr$^{-1}$ (5.4 Tg P yr$^{-1}$) are added to the coastal oceans as reactive P (Figure 2). This is close to the amount estimated by Föllmi (1995), and within the range of other literature values for

**Table 1.** Overview of simulations and riverine nutrient fluxes:

| Simulation | N-flux in Tmol N yr$^{-1}$ | P-flux in Tmol P yr$^{-1}$ | Description |
|---|---|---|---|
| CTR | 0 | 0 | UVic simulation without NEWS (Control) |
| NEWS-N | 1.6 | 0 | UVic simulation with DIN from NEWS |
| NEWS-P | 0 | 0.17 | UVic simulation with P from NEWS, no burial |
| NEWS-N+P | 1.6 | 0.17 | UVic simulation with N and P from NEWS, no burial |
| N+P-BURLOW | 1.6 | 0.17 | UVic simulation with N and P from NEWS, low burial configuration: $Y_F = 100.5$; $A = 90$; $r = 38$. (Kemena et al., 2019) |
| N+P-BURHIGH | 1.6 | 0.17 | UVic simulation with N and P from NEWS, high burial configuration: $Y_F = 123$; $A = 112$; $r = 32$ (Flögel et al., 2011) |

global fluvial fluxes of bioavailable P (Kemena et al., 2019; Ruttenberg, 2003; Benitez-Nelson, 2000; Compton et al., 2000). In NEWS-P and NEWS-N+P no sink of P was implemented, and these simulations are characterized by continually increasing P inventory (Figure 3a). Benthic burial fluxes of P amount to 0.16 Tmol yr$^{-1}$ in N+P-BURLOW and 0.24 Tmol yr$^{-1}$ in N+P-BURHIGH, leading to an imbalance of +0.01 Tmol yr$^{-1}$ and -0.07 Tmol yr$^{-1}$ respectively.

These results range within the estimates for burial rates from observations, which globally vary around 0.2 Tmol P yr$^{-1}$: 0.11–0.34 Tmol P yr$^{-1}$ in Benitez-Nelson (2000), 0.17-0.24 Tmol P yr$^{-1}$ for Ruttenberg (2003) and 0.21 Tmol P yr$^{-1}$ in Filipelli (2008).

The highest simulated P burial fluxes can be found in coastal regions, especially in the western Pacific and Atlantic ocean (Figure A2). Simulated burial hot spots are situated in proximity to the coasts of China, East Russia and Alaska, next to the eastern coast of North- and South-America and in the North Sea. There are no global observational data sets of P burial. Nevertheless, the distribution of the benthic fluxes is similar to other model studies that have used a similar algorithm (e.g. Bohlen et al., 2012).

The inclusion of riverine P has a significant impact on global P concentrations. In simulation CTR without riverine nutrient supply, surface phosphate concentrations range from 0 mmol P m$^{-3}$ in most of the western tropical and subtropical ocean basins to more than 2 mmol P m$^{-3}$ in the Southern Ocean (Figure 4). The addition of riverine N (in NEWS-N) leads only to small changes in the surface P concentrations, in particular P declines reflecting enhanced biological processes (P uptake) in the coastal oceans (not explicitly shown here). The regions with a decrease in P can be found in the shelf and coastal oceans, corresponding to the regions, where N concentrations increase in Tivig et al. (2021). Nevertheless, the magnitude of P changes is small compared to the spatial variance of surface phosphate concentrations of 0.64 mmol P m$^{-3}$ in CTR and 0.85 mmol P m$^{-3}$ after 10,000 years of riverine N supply in NEWS-N (0.65 mmol P m$^{-3}$ at the start of the simulation). For simulations with inclusion of riverine P but without benthic burial of P, surface P concentrations after 10,000 years of riverine P supply are

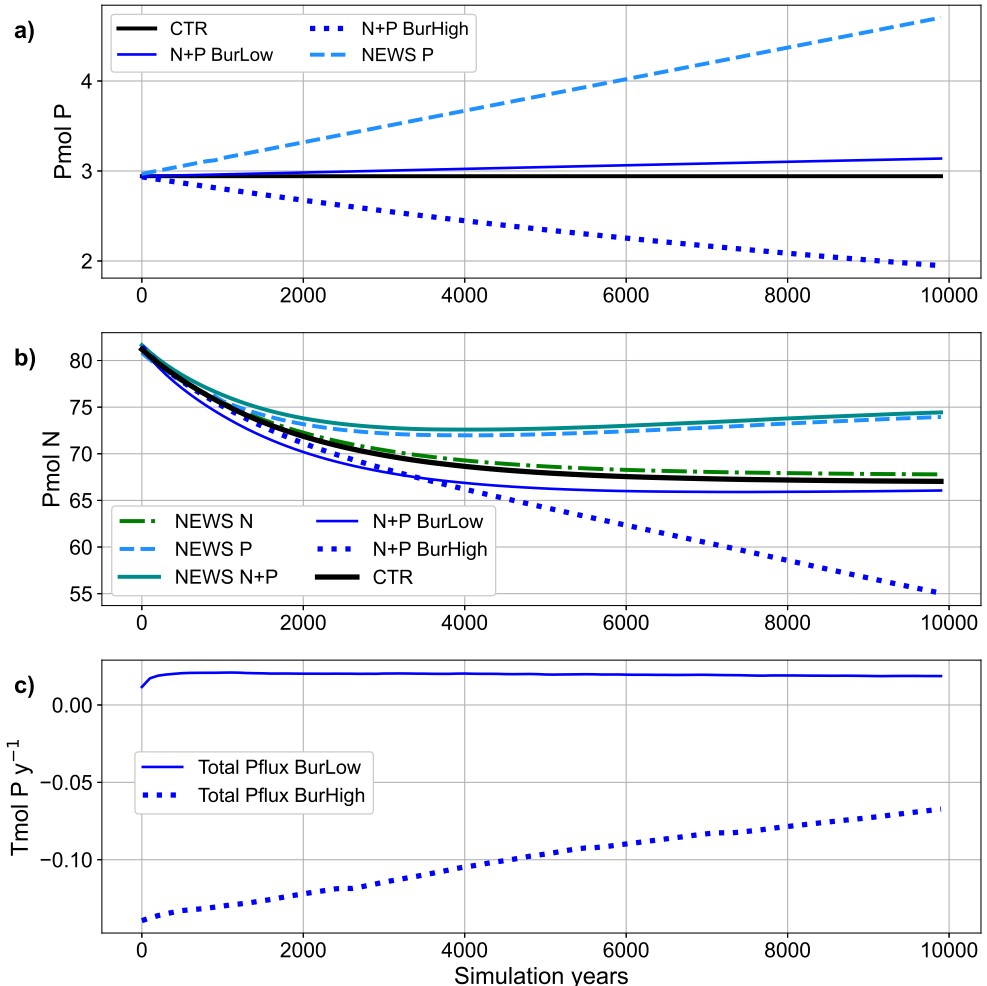

**Figure 3.** a) Timeseries of global phosphorus in all simulations over the 10,000 simulation years in Pmol P. b) Timeseries of global nitrogen in all simulations over the 10,000 simulation years in Pmol N. c) Timeseries of total phosphorus fluxes (sum of riverine input and burial flux) in Tmol P per year for the simulations with low and high burial. Simulation descriptions can be found in Table 1.

much higher than in the control simulation (Figure 4 c, d.), independent of the additional riverine N supply. Additional P is distributed over the global oceans except for the tropical Atlantic and the north-western tropical Pacific Ocean. These regions correspond to the northern subtropical gyres and are known to be oligotrophic and phosphate limited (Mather et al., 2008;

Martiny et al., 2019). The other extreme is the simulation N+P-BURHIGH, where the modelled ocean is loosing P in all those regions, where surface P concentration is different from 0 in CTR (Figure 4 f.). Only moderate changes and a spatial variance of 0.69 mmol P m$^{-3}$ are the result of the simulation with riverine N and P and low burial functions (N+P-BURLOW, Figure 4 e.).

After 10,000 years of simulation, of all experiments including riverine phosphate, the global average P concentration in N+P-BURLOW is nearest to observed present-day oceanic conditions (observations with data from the World Ocean Atlas (Garcia et al., 2019b) are shown in Figure A4). Nevertheless, each scenario provides a different insight in marine biogeochemical feedbacks.

## 3.2  P influence on oceanic N inventory and N-cycle feedbacks

The addition of riverine P to the modelled ocean has a considerable impact on the distribution of simulated NO$_3$. At the end of CTR, NEWS-N and the simulation N+P-BURLOW, the N sink by denitrification and the N sources by N$_2$-fixation and riverine input are balanced (Figure 3, Figure 5). This is not the case for NEWS-P and NEWS-N+P, where the oceanic N inventory slightly increases even at the end of the simulations, and is very different from N+P-BURHIGH, where N concentrations decrease continuously. This evolution is also different from the P inventory, with its asymptotic shape (Figure 3). In NEWS-P, where only P is added via river runoff, global oceanic N is only slightly lower than in NEWS-N+P and considerably higher (around 6 Pmol N) than in NEWS-N. This is mainly the result of the increase in N$_2$-fixation triggered by the additional P flux (Figure 5). Comparing NEWS-N, NEWS-P, and N+P-BURLOW shows, that while global benthic denitrification stays nearly constant during the simulations, water column denitrification and N$_2$-fixation develop differently depending on the experimental design (Figure 5): additional P leads to an increase in global denitrification and N$_2$-fixation. Including low burial functions stabilizes both fluxes over time, but at higher levels than in NEWS-N. In N+P-BURHIGH N$_2$-fixation and water column denitrification decrease significantly over the simulations, but seam to stabilize at the end of the 10,000 years. However, the flux of benthic denitrification decreases slower and has not stabilized at the end of the experiment, causing the continuous decrease in N in this simulation. To see a stabilized budget here, the simulations should have been much longer.

In the upper oceans, NO$_3$ concentrations increase with supplemental P, except near the main oxygen minimum zones (OMZ) of the Gulf of Guinea, the northern Indian Ocean and the tropical eastern Pacific. Without the burial of P, N increases especially in the tropical Pacific, the North Atlantic and higher latitudes, but decreases in the tropical eastern Pacific. Including P burial leads to a smaller increase in N and even a high decrease in N in the simulation with high P burial (Figure 6). Including riverine P has more impact than including riverine N supply alone. If more P is buried, N concentrations decrease globally. The above pattern of N shows increasing concentrations almost everywhere in experiments NEWS-P and NEWS-N+P compared to CTR, but is almost reversed for experiment N+P-BURHIGH, with N concentrations increasing only near the OMZ in the Bay of Bengal and off the Pacific coast of Central-America (Figure 6).

These changes in NO$_3$ concentrations are the result of P fluxes impacting the two main processes of the N feedback cycle, denitrification (Figure 9) and N$_2$-fixation (Figure 8). It is widely assumed, that the surface NO$_3$ to PO$_4$ ratio is a dominant

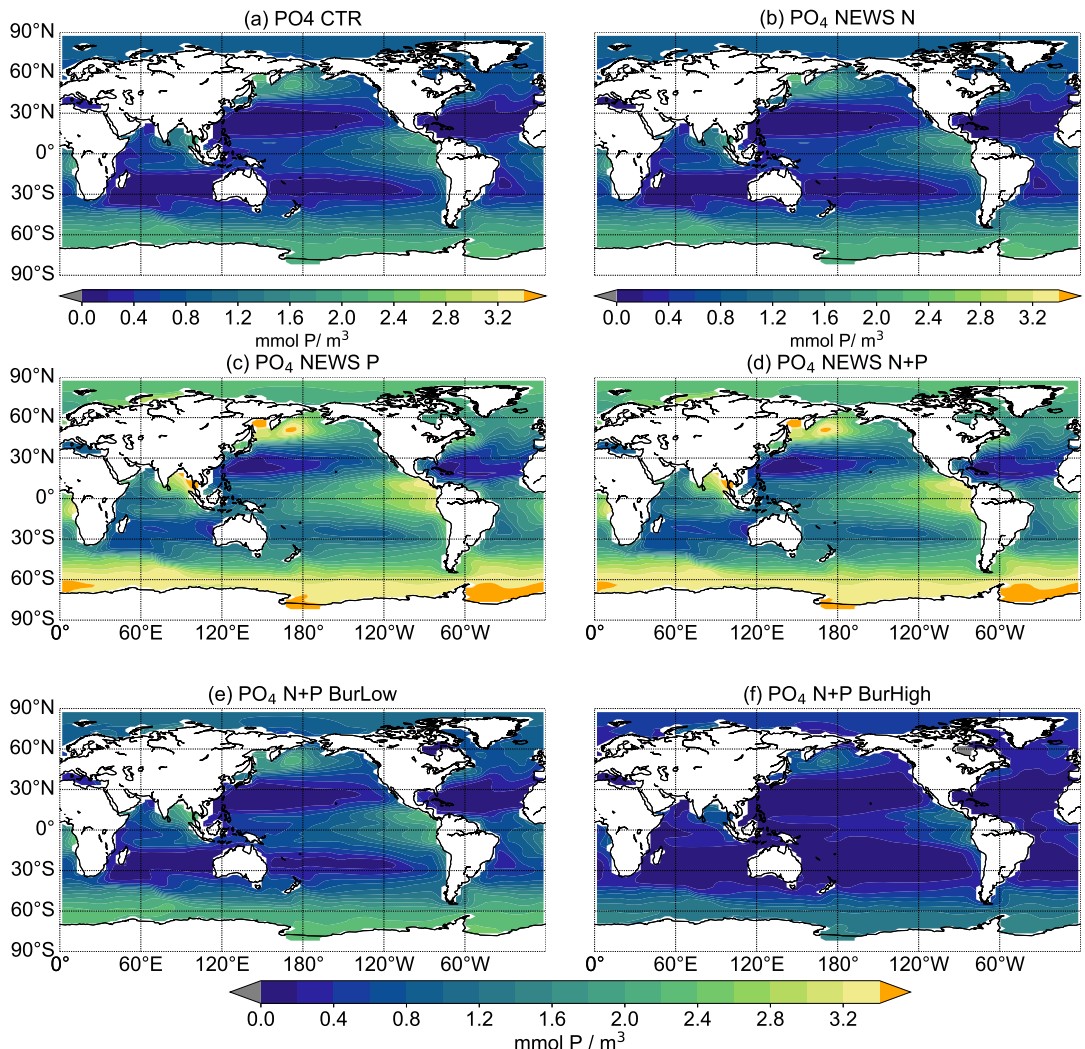

**Figure 4.** Global distribution of $PO_4$ concentrations in mmol P m$^{-3}$ averaged over the upper 180 m at the end of the respective simulations over 10,000 years. (a) Global distribution of $PO_4$ concentrations in CTR (b) NEWS-N (c), NEWS-P (d), NEWS N+P (d) N+P-BURLOW (e) and N+P-BURHIGH (f). The orange colour indicates regions, where the $PO_4$ concentrations exceed 3.4 mmol P m$^{-3}$.

controlling factor of these feedback mechanisms (Gruber, 2008). Diazotrophs are limited by P, especially in regions, where light, temperature and iron availability are not limiting. The regions where $N_2$-fixation is enhanced in the simulations with

riverine P (NEWS-P, NEWS-N+P and N+P-BURLOW) corresponds well with regions of general P limitation (compare with (Kemena et al., 2019), Figure 8). $N_2$-fixation is stimulated here by the addition of P to the ocean and is generally sensitive to changes in P supply. In the experiment with high burial rates, these same regions are charaterized by decreasing $N_2$-fixation

rates (Figure 8, (f)).

Denitrification is the other feedback mechanism controlling the global ocean N budget. In the simulations where N concentrations increase due to higher $N_2$-fixation rates, denitrification is also enhanced. However, in the simulation with high burial rates of P, denitrification rates are much lower than in the control simulations in the Gulf of Guinea, the Bay of Bengal and the eastern tropical Pacific (Figure 9, (f)), which also leads to higher concentrations of N in these regions (Figure 6). Both processes, $N_2$-fixation and denitrification, are increased in the simulation with additional riverine P. Where they take place in separated regions of the tropical and subtropical oceans, these processes lead to an increase in N in the global surface ocean ($N_2$-fixation) and a loss in N in the water columns near the main OMZ of the ocean (denitrification). This is not the case when $N_2$-fixation and denitrification are co-located for example in the Bay of Bengal.

In our previous study with the addition of only riverine N, the "vicious cycle" defined by Landolfi et al. (2013) was suspected to be the reason for a significant decrease in N concentration in the Bay of Bengal, even though rivers exported additional N to the sea (Tivig et al., 2021). The addition of riverine P improves this process here, so that the ocean is loosing N not only in the Bay of Bengal, but also in the upper tropical Atlantic ocean basin (Figure 7d,g,j,m) near the Gulf of Guinea and in the upper tropical Pacific ocean (Figure 7e,h,k,n).

The main driver for this vigorous vicious cycle in these simulations is the supply of P from rivers and the interplay of different feedback loops in marine biogeochemistry (Figure 10). Considering N on the one side, additional N from rivers or atmospheric fixation enhances marine biological production, which contributes to higher export rates, leading to remineralization and oxygen consumption. In regions where oxygen is scarce, denitrification will occur and consume N. $N_2$-fixation is limited by P. Therefore, adding more P, for example, from rivers, can enhance this process and by this the vicious cycle. On the other hand, increasing P can also lead to lower oxygen concentrations and therefore more denitrification.

Note that iron availability also plays an important role. It is a limiting factor of diazotroph growth and hence $N_2$-fixation (Landolfi et al., 2013). Iron availability can lead to a shift in the growth of N fixers to regions with higher iron availability, and therefore also spatially decouple denitrification and $N_2$-fixation, breaking the vicious cycle (Landolfi et al., 2013; Moore et al., 2009). Our UVic model represents iron including a static concentration mask. Therefore no interactive response to perturbations of ocean biogeochemistry are possible and iron availability and limitation will not change during the simulations.

### 3.3 Phosphorus cycling and oxygen minimum zones

Low ventilation of the water column and/or high rates of remineralization of organic matter can cause oxygen depleted waters. Niemeyer et al. (2017) found that an increase in the marine phosphorus inventory under assumed business as usual global warming conditions could lead to a 4- to 5-fold expansion of the suboxic water volume over millennial timescales. Several studies also suggest, that $O_2$ depletion in coastal regions caused by eutrophication may enhance the release of P from sediments, thus providing additional P (Flögel et al., 2011; Wallmann, 2010; Ingall and Jahnke, 1994). These two processes together form a positive feedback loop, enhancing oxygen depletion and expansion of the oxygen minimum zones (Oschlies et al., 2018) (Figure 10). The processes relevant for this feedback are all included in the burial experiments.

Three regions, the Gulf of Guinea, the Bay of Bengal and the eastern tropical Pacific ocean, are characterised by low to very low oxygen concentrations (Figure 11a).

The black contour lines in Figure 11 indicate the extent of the oxygen minimum zones at 300 m depth, averaged over the last 100 years of the simulations. The oxygen minimum zone is defined here by $O_2$ concentrations lower than 70 mmol m$^{-3}$. The main regions with low oxygen concentrations are known to be situated in subsurface waters of the Arabian Sea and in the areas of the eastern boundary upwelling regions in the tropical oceans off California, Peru and Namibia (e.g. Oschlies et al., 2018). The model results (Figure 11(a)) show, that UVic misplaces the oxygen minimum zone in the Indian Ocean from the Arabian Sea to the the Bay of Bengal. This is similar to other biogeochemical ocean models (Séférian et al., 2020). In reality, the Bay of Bengal is a region with strong seasonality driven by the Asian monsoon system (Löscher et al., 2020). Therefore highly variable oxygen concentrations inhibit denitrification while high water column denitrification has been observed in the Arabian Sea (Johnson et al., 2019; Bange et al., 2005).

In the simulations with increased P from rivers, oxygen concentrations decrease significantly in all tropical oceans as well as in the eastern boundary regions. P burial damps the oxygen depletion and can even lead to an increase in oxygen concentrations especially in the tropical oceans and in the OMZs (Figure 11). Changes in oxygen concentrations are not limited to oxygen minimum zones and regions with changing oxygen concentration do not exactly correlate with regions of changed denitrification rates (Figure 9).

An observational estimate of today's suboxic water area and volume equals $30.4 \pm 3$ millions of km$^2$ and $102 \pm 15$ millions of km$^3$ respectively, for oxygen concentrations less than 20 mmol m$^{-3}$ (Paulmier and Ruiz-Pino, 2009). In CTR, OMZ defined as regions with oxygen concentration less than 20 mmol m$^{-3}$ (less than 70 mmol m$^{-3}$) cover 13.1 millions of km$^3$ (52 millions of km$^3$). The addition of riverine N alone leads only to small changes in oxygen concentrations with a volume of OMZ of 13.2 (54) millions of km$^3$. Adding P leads to a strong increase in OMZ area with up to a 68 (192) millions of km$^3$ expansion in NEWS-N+P. With decreased P concentrations, $O_2$ concentrations increase globally and the global volume of suboxic waters is reduced to less than 1 (5) millions of km$^3$. The switch for water column denitrification is mainly controlled by the oxygen concentration (Gruber, 2008). While adding riverine N does not impact ocean denitrification significantly, the addition of P leads to an increase in denitrification in the three main regions with oxygen deficit waters (Figure 12). Consequently this leads also to reduced N concentrations in these same regions. In contrast, high P burial rates worldwide and hence lower P concentrations lead to significantly higher $O_2$ concentrations, decreased rates of denitrification and therefore higher N concentrations. This is especially the case in regions, where the N cycle feedbacks had previously limited higher N concentrations (Figure 13 (f)). Including riverine N supply only slightly impacts the $O_2$ distribution, but can start a negative feedback loop near oxygen minimum zones. In our first experiment this phenomenon also impacted primary production in these regions, damping a global increase in marine production compared to the control simulation. Adding P from riverine export to the modelled ocean has a higher impact on N and oxygen concentrations. As $O_2$ plays an important role in the N feedback cycles, the impact is more significant and not only limited to the OMZ (refer also to Figure 10).

## 3.4 Primary Production

In Tivig et al. (2021) we showed that including riverine N had only limited impact on marine productivity, due at least partially to feedback reactions in the marine N cycle. Additional P significantly changes this result (Figure 14): Comparing the simulation with riverine N only with the simulation with riverine P only shows that, at least in our model and on a millennial timescale, P is more limiting for primary production than N alone. Primary production amounts to 67 Pg C yr$^{-1}$ globally in NEWS-P compared to only 55 Pg C yr$^{-1}$ in NEWS-N. Even in the low burial simulation, marine biology is more productive than in the simulation without riverine P addition (59 Tg C yr$^{-1}$). In N+P-BURHIGH, on the other hand, where the ocean is deprived of P at the end of the simulation, the production rates decrease in the global tropical oceans to reach a global integral 33 Pg C yr$^{-1}$. Only near the river mouths, where burial has not yet been effective, the NPP rates are higher than in the control simulation. In summary, NPP rates are sensitive to the addition of P in our model. In the simulations with N and P addition from rivers, the feedbacks are still active, but NPP rates are nevertheless higher than in our simulation with riverine N alone.

## 4 Discussion and limitations

In this study we compare simulations with different biogeochemical settings for N and P in the same ocean circulation. We find that the addition of riverine phosphorus to coastal oceans in our model has a greater impact on the nitrogen cycle and its internal feedbacks than the addition of N alone (Figure 10). First, additional P directly impacts the N budget by fostering N fixation, especially in P-limited regions. Secondly, it fuels marine primary production, which leads to an increase in the export of detritus, and hence remineralisation, which in turn reduces oxygen levels. Lower oxygen concentrations increase denitrification and work towards balancing the nitrogen budget globally. We have also shown that there are regional patterns and that locally, especially in proximity to regions with low oxygen concentrations, the addition of riverine P can lead to depletion of N as a consequence of a positive feedback loop.

Nevertheless, our experimental settings have some limitations that need to be accounted for. First, sedimentary N-loss is simulated according to an empirical transfer function based on organic carbon sinking flux to the sediments and bottom-water dissolved oxygen and nitrate. We apply the empirical function of Bohlen et al. (2012). Since our coarse resolution model does not fully resolve narrow continental shelves and coastal dynamics, a subgrid-scale bathymetry parameterization is included (Somes and Oschlies, 2015). However, UVic still underestimates the sedimentary loss of N in continental shelves (see also Somes et al. (2017)).

Although the coastal ocean only accounts for a small part of the total oceanic area, it plays an important role for biogeochemical cycles and its contribution to biogeochemical fluxes is disproportionately large (Naqvi and S.Unnikrishnan, 2009). The coarse resolution of coastal regions in UVic, with no explicit simulation of the coastal processes, might therefore lead to an underestimation of these processes. In another modelling experiment Rabouille et al. (2001) found for example, that enhanced riverine nutrient export leads to a substantial increase in primary production in the coastal ocean and hence accumulation of biomass in all compartments of the coastal ocean system, while denitrification did not increase proportionally in their experiment. These results suggest that the coastal ocean cannot generally self-regulate the effects of perturbations on nutrient cycles (Rabouille

et al., 2001). In our experiment, additional N and P from river export could therefore lead to regional shifts in the patterns of primary production and denitrification compared to a model with higher resolution of the coastal oceans. If denitrification rates are too high in our model setup, the vicious cycle might also be overestimated. Furthermore, because nutrient retention on the coastal shelf is not included in our model setup, the buffer effects of the coasts cannot fully be taken into account. Riverine nutrient export to the open ocean might therefore be overestimated in our experiments. These processes could be parametrized as in Sharples et al. (2017) or Izett and Fennel (2018) in a future experiment.

Additionally, submarine groundwater discharge to the oceans is not simulated, but might be another significant component of biogeochemical budgets (Santos et al., 2021; Slomp and Capellen, 2004).

The current study does not incorporate all feedbacks that can potentially impact the N cycle. For instance, changes in nutrient availability have been found to affect the C:N:P ratio of primary producers, both through physiological acclimation and shifts in species composition (Grosse et al., 2017). This has then in return an effect on nutrient cycling and hence primary production. It would be interesting to repeat (some of) the simulations with the optimality-based flexible stoichiometry model that has recently been coupled to the UVic model (Chien et al., 2023) and revealed that the same feedback processes reported here also operate for a large range of physiological and climatic boundary conditions, but that there is a tendency for quantitatively weaker responses compared to the tight coupling found in fixed-stoichiometry models such as used here.

Because coastal waters are generally iron replete, we do not expect substantial near-field impacts of iron-cycle feedbacks in coastal regions. Further offshore, and particularly in currently iron-limited regions such as the Southern Ocean and the subpolar North Pacific, the feedbacks studied here could be modified by changes in atmospheric iron supply, as investigated by earlier studies (Krishnamurthy et al., 2009; Jickells and Moore, 2015; Giraud et al., 2008).

Also, the spatial patterns of NPP and $N_2$-fixation differ from observations. In the Indian Ocean, the model simulates too much $N_2$-fixation in the Bay of Bengal and not enough in the Arabian Sea, a common problem of coarser resolved models, as already stated earlier (Séférian et al., 2020; Moore and Doney, 2007). Of the three major quasi-permanent sites of water column denitrification (Arabian Sea, the Eastern tropical North Pacific and the Eastern tropical South Pacific off the coast of Chile (Codispoti, 2007)), the model is only able to reasonably simulate the eastern tropical North Pacific one. In the Eastern tropical South Pacific, modelled denitrification is too close to the equator and does not occur off the coast of Chile where it should (Keller et al., 2012).

Nevertheless, the UVic model has been tested and used for several previous studies of marine biogeochemistry. Keller et al. (2012) found that surface nitrate and phosphate concentrations are similar to observational data (Garcia et al., 2019a). The model rates of $N_2$-fixation are within the wide range ($\sim$ 7-15 Tmol N yr-1) of global $N_2$-fixation estimates (Codispoti, 2007; Deutsch et al., 2007; Gruber and Sarmiento, 1997; Karl et al., 2002). Despite the model limitations mentioned before, the University of Victoria Earth System Climate Model of intermediate complexity has been proven useful to investigate long-term changes in marine biogeochemistry (e.g. Mengis et al., 2020; Somes et al., 2017; Keller et al., 2012).

For the riverine nutrient input, we used the results of the NEWS2 model. Mayorga et al. (2010) and Dumont et al. (2005) evaluated the individual models for the river export and found, that despite uncertainties associated with model inputs, nutrient yield and export were close to observations and the global export estimates were similar to the results of previous analyses.

Although the riverine nutrient export from NEWS2 includes the anthropogenic component, based on the year 2000, we did not focus on the anthropogenic component in our experiment and the millennial-scale simulations do not provide much insight into the short time variability. Nevertheless, the question of the influence of human activities on the marine N cycle is most central. Human activities have markedly altered the earth's cycles of the nutrients nitrogen and phophorus. Beusen and Bouwman (2022) have shown that rivers with a nutrient load that is more than 50 % and at the same time elevated N:P ratios (> 25) contributed 36 % to the total global N export to coastal waters. They also found, that from 1970 to 2015, global N and P delivery to surface waters has increased from 44 to 71 Tg N yr$^{-1}$ for N (3.1 to 5.1 Tmol N yr$^{-1}$) and 7.1 to 9.7 Tg P yr$^{-1}$ (0.23 to 0.31 Tmol P yr$^{-1}$) for P, although the export has decreased in industrialized countries. Changes do not only affect the absolute amount of nutrients, but also the N:P ratios. Understanding the interplay between these two nutrients is therefore important, in order to predict changes in coastal and global oceans (Beusen and Bouwman, 2022).

## 5  Conclusions

In a previous study, Tivig et al. (2021) added a new component to the global Earth system model UVic, to simulate the effects of N supply from river discharge. They found that internal feedbacks in the nitrogen cycle mostly compensated for the imposed yearly addition of 1.6 to 3.3 Tmol of riverine nitrogen and limited the impact on global marine productivity to <2 %. In the current study, riverine phosphorus has been included separately and in addition to N. After 10,000 years, only a simulation with low P burial rates reached a steady state. After 10,000 years of simulations, the global amount of P in the ocean has increased significantly in the simulations where riverine P is not balanced by burial sinks and continues to increase. In the simulation with low burial rates, global P increases only slightly, while the simulation with high burial rates leads to a continual loss in global P. The simulations showed that, on millennial timescales, including riverine P in the model has a greater impact on the modelled marine biogeochemistry and biology than the inclusion of N alone. Therefore, we can answer the questions raised at the start:

– Additional P from river runoff affects marine biology not only near the river mouths but also in regions far off the coasts and in the deep oceans.

– Because the additional P (as well as the loss of P via burial) affects the two processes denitrification and N$_2$-fixation, the N-cycle is also affected and N concentrations increase or decreases locally and globally. In simulations where P is added without burial, N concentrations also increase globally, except in regions where denitrification is dominant, in proximity to regions with low to very low oxygen concentrations. Balancing the addition of P by P burial leads to a decrease in the global marine N concentration. The global amount of N at the end of the 10,000 years is lower in the burial simulations than in the simulations without P compensation. This can be attributed to internal feedbacks in the N cycle: additional P increases N$_2$-fixation but also denitrification. Knowing that denitrification consumes 7 mol of NO$_3$ for every mole of organic N provided by N$_2$-fixation, in regions with low oxygen concentration, a 'vicious cycle' leads to a runaway loss of N. In the simulations where P is not limiting, the N loss can be globally compensated by N$_2$-fixation in other regions.

Where P is reduced because of sedimental burial, even if only low, $N_2$-fixation is not effective enough to compensate for the loss in N due to denitrification.

– Adding P in the coastal oceans has significant impact on marine oxygen concentrations. Simulated oxygen concentrations decrease globally but strongest in the tropical and Nordic oceans, if P is added without an additional sink. In the simulation where P is lost due to sediment burial, however, $O_2$ concentrations increase globally. In Tivig et al. (2021) we found that $O_2$ concentrations are only slightly impacted by the addition of riverine N alone.

Finally, our study showed that additional P from riverine input strongly influences marine productivity not only in the coastal oceans, but also in the open oceans worldwide. Our model simulations suggest that, on millennial timescales, the impact of riverine P on ocean biogeochemistry is more important than the one of riverine N. While this result is linked to our spatially resolved model configuration with all its limitations described before, it confirms the conclusion of Tyrrell (1999) derived for a box model. In our experiments we also showed that internal feedbacks in the N and P cycle play a crucial role for marine biogeochemistry and can highly impact regional biology, especially in the coastal oceans.

The main sources of P as the "ultimately limiting nutrient" are rivers transporting P provided by weathering and human activities (Giraud et al., 2008; Föllmi, 1995). While weathering processes are active on millennial timescales and therefore only slowly change the amount of P advected by rivers, human activities have influenced these nutrient transports more rapidly during the last centuries (Beusen et al., 2016). Furthermore, changes in atmospheric carbon dioxide concentrations and resulting climate changes have already, and will even more so in the future, impact riverine export as well as ocean biogeochemistry (Gao et al., 2023). Although the uncertainties in the real and the modelled nutrient fluxes are still large, our simulations suggest that global ocean biogeochemistry can be substantially affected by the supply of nutrients from rivers and that the global representation of biological activity may be improved by considering riverine export and coastal processes. For this purpose, a better spatial resolution of the coastal oceans as well as a more realistic representation of coastal N and P cycling processes could be helpful, also in order to analyse the variability on shorter timescales.

*Code and data availability.* The data and material that support the findings of this study are available through GEOMAR at https://hdl.handle.net/20.500.12085/85adfcd5-bc86-440c-a205-496749a9025f (Tivig et al., 2024). More information on the original NEWS2 data set is available from the Global NEWS group at the web site http://icr.ioc-unesco.org/index.php. Please email Emilio Mayorga at mayorga@marine.rutgers.edu to obtain this data set.

**Appendix A:  Supplemental Figures**

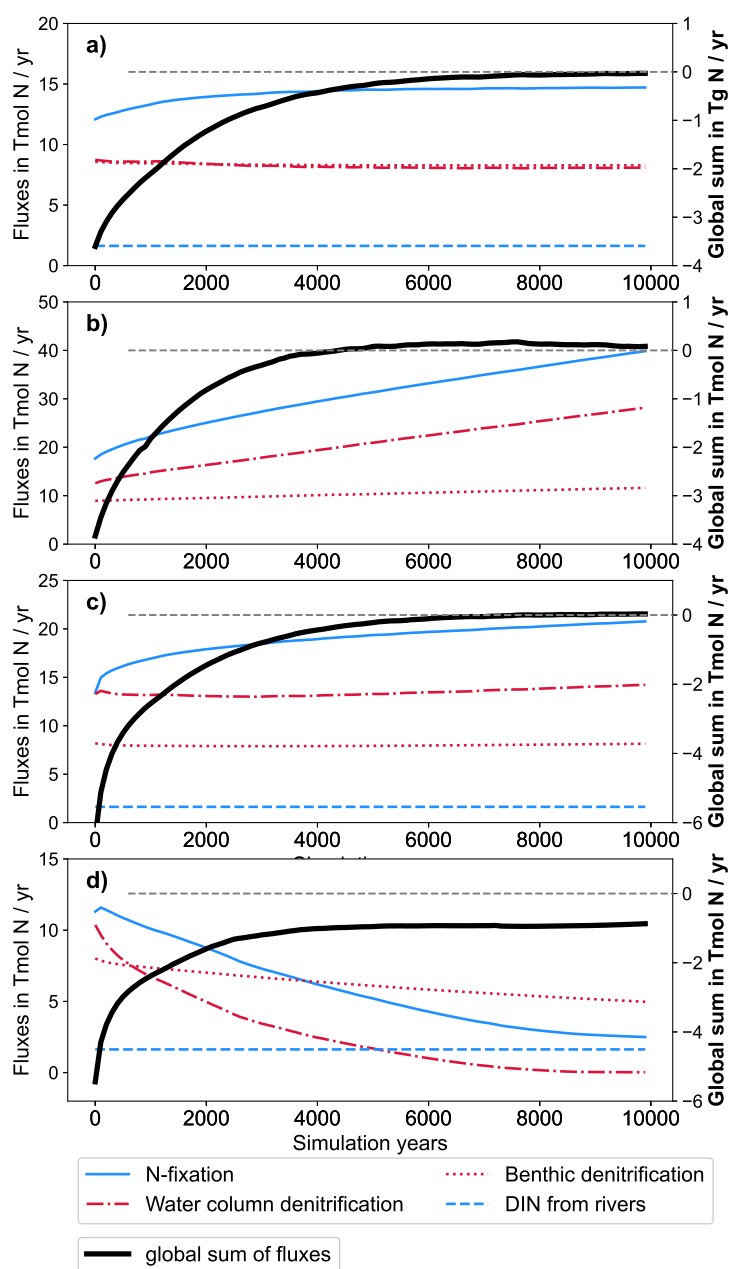

**Figure 5.** Timeseries of global nitrogen fluxes from NEWS-N (a), NEWS-P (b), N+P-BURLOW (c) and N+P-BURHIGH (d) over the 10,000 simulation years in Tmol N per year. Nitrogen fixation (blue solid line) and riverine N input (blue dotted line) are balanced by water column denitrification and benthic denitrification (red solid and red dotted line, respectively). The global sum of all N fluxes is shown as bold black line. Fluxes are given in absolute values. Note the different scales of the y-axis in the different panels. The dotted line in grey is the zero-line of the global sum of N fluxes.

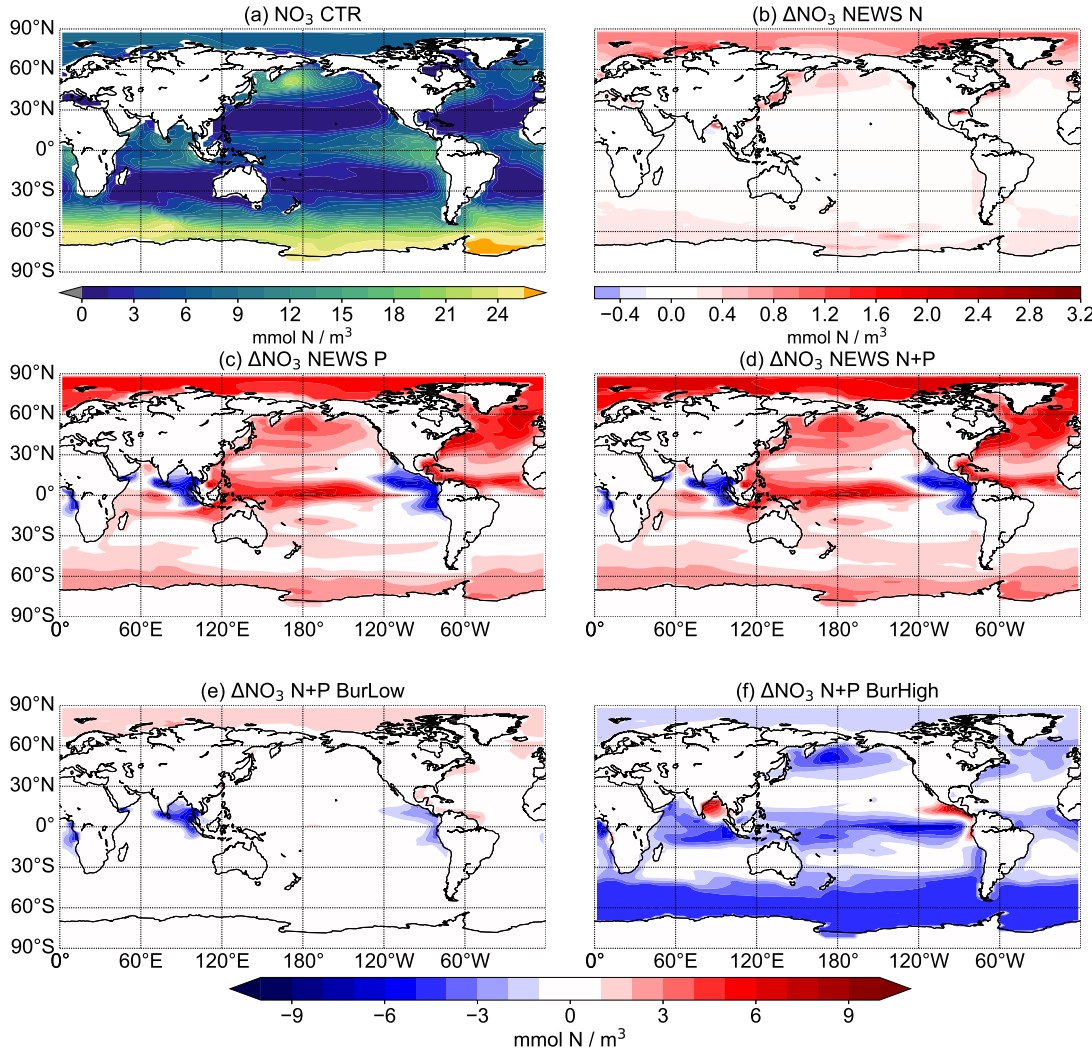

**Figure 6.** Global distribution of $NO_3$ concentrations averaged over the upper 180 m in mmol N m$^{-3}$. The panel (a) shows $NO_3$ concentrations in CTR. Panels (b)-(f) show N concentrations in the simulations NEWS-N, NEWS-P, NEWS-N+P, N+P-BURLOW and N+P-BURHIGH as difference to CTR.

Δ NO$_3$

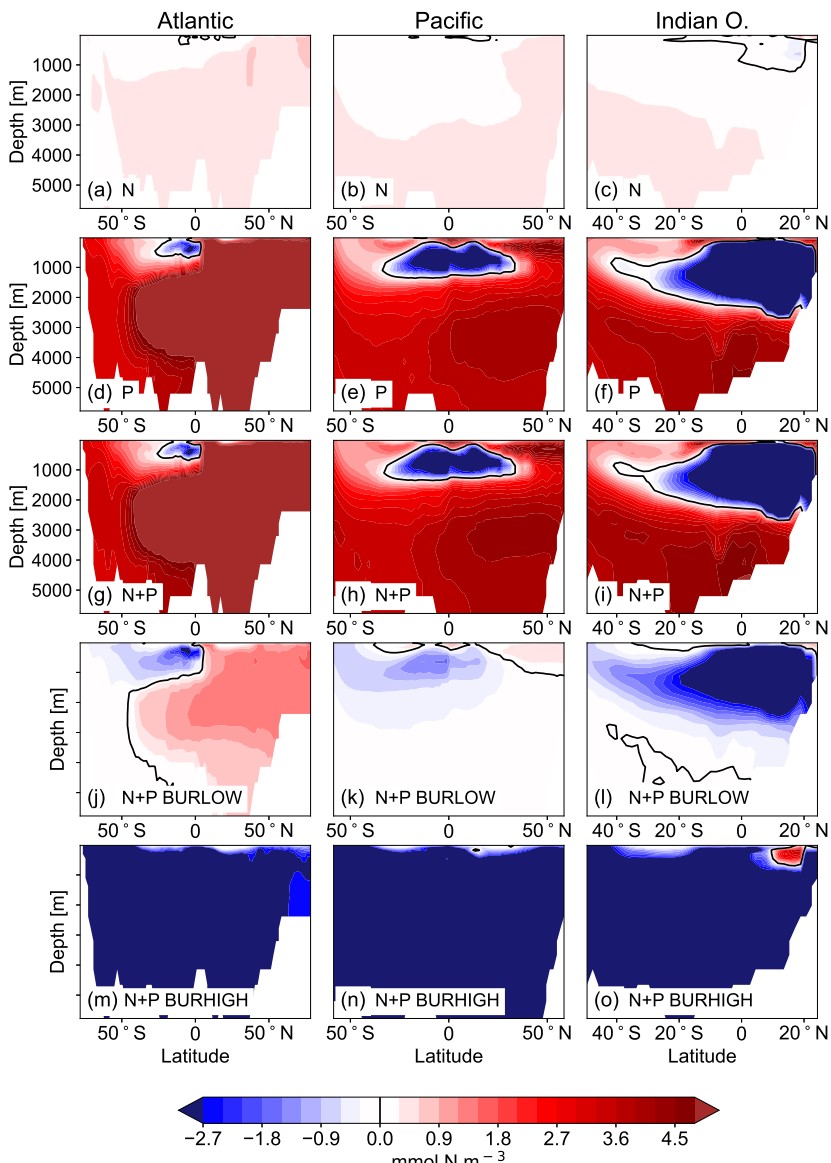

**Figure 7.** Difference in zonal mean concentrations of NO$_3$ in the main ocean basins (Atlantic: left column, Pacific: middle, and Indian Ocean: right) in mmol N m$^{-3}$ as difference between the simulations NEWS-N, NEWS-P, NEWS-N+P, N+P-BURLOW and N+P-BURHIGH and CTR.

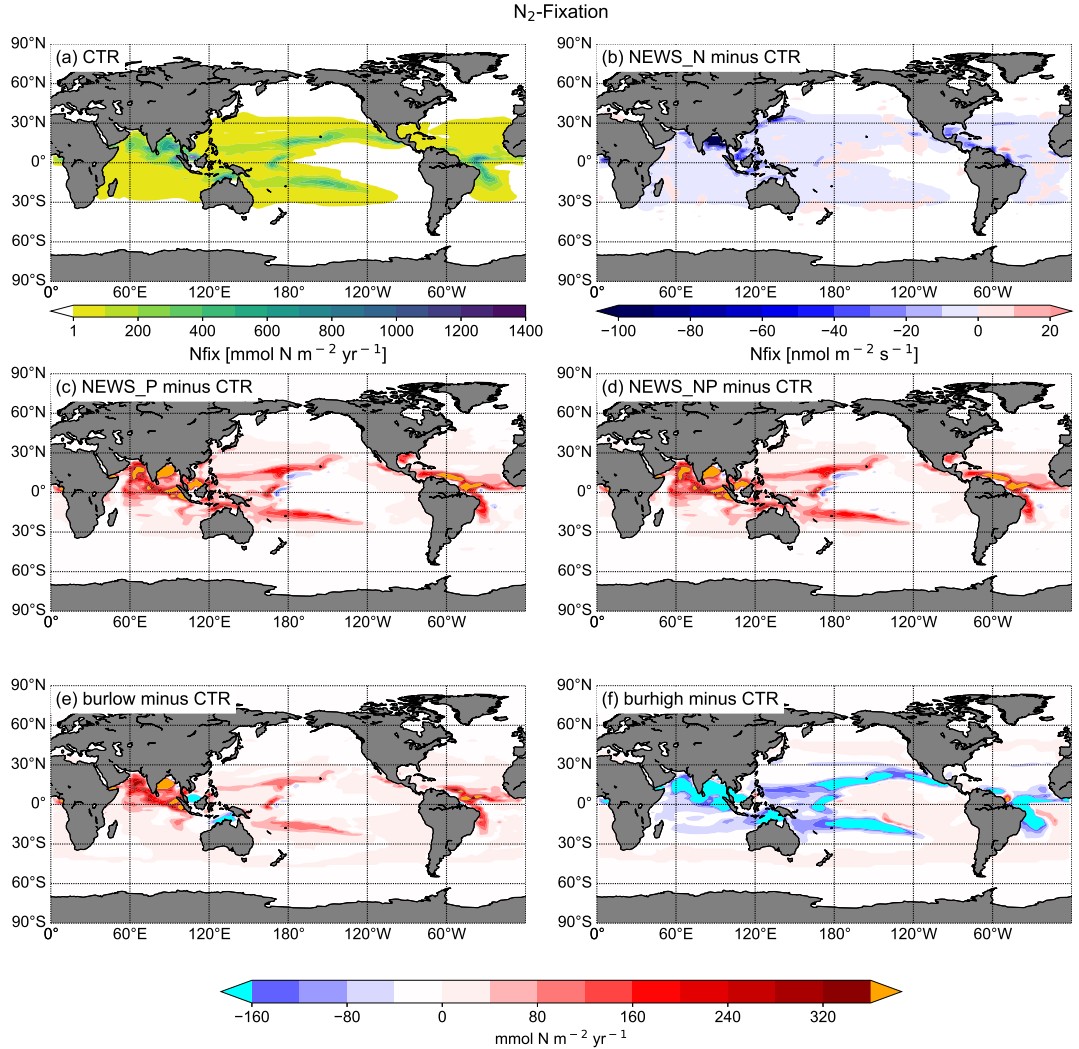

**Figure 8.** Vertical integration of $N_2$-fixation in mmol N m$^{-2}$ yr$^{-1}$. (a) $N_2$-fixation in CTR. (b)-(f) difference in $N_2$-fixation between the simulations NEWS-N, NEWS-P, NEWS-N+P, N+P-BURLOW and N+P-BURHIGH and CTR. The cyan colour indicates regions with very low $N_2$-fixation rates compared to CTR. The orange colour indicates regions with very high $N_2$-fixation rates compared to CTR.

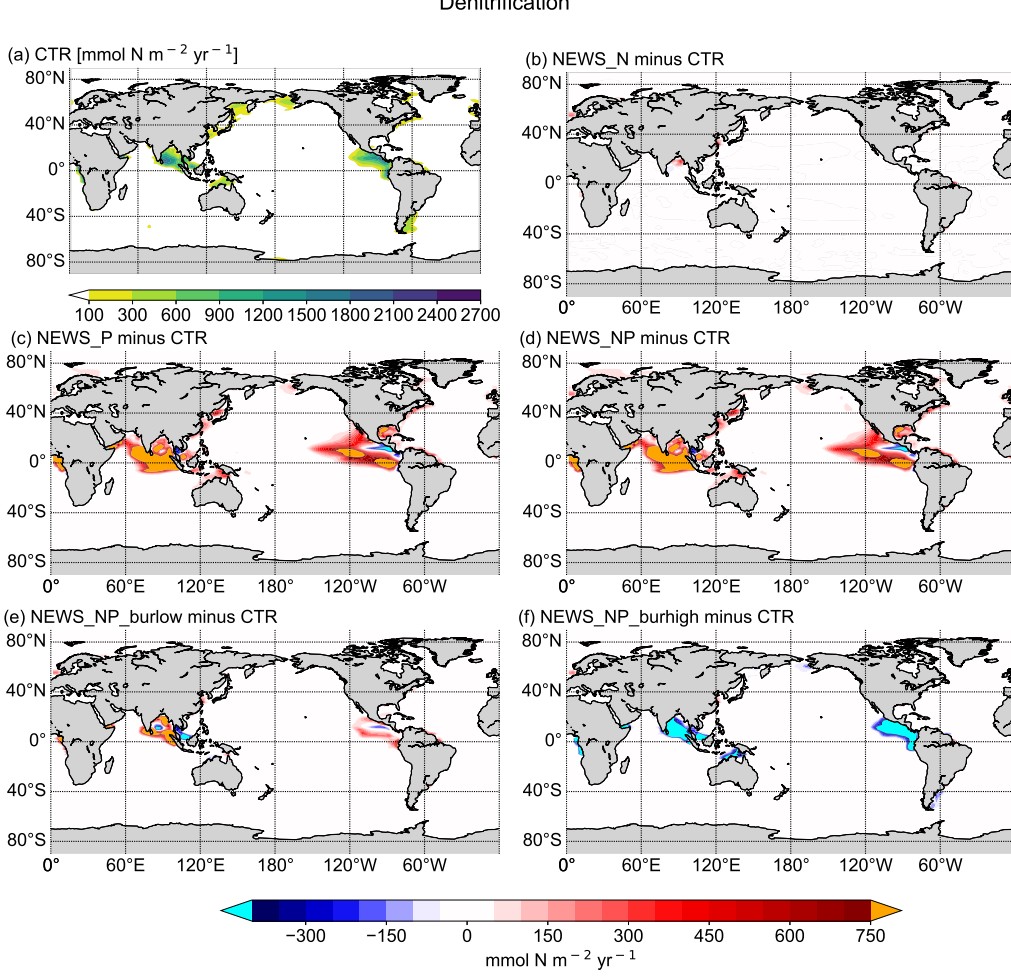

**Figure 9.** Vertical integration of denitrification in mmol N m$^{-2}$ yr$^{-1}$. (a) Denitrification rates in CTR. (b)-(f) difference in denitrification rates between the simulations NEWS-N, NEWS-P, NEWS-N+P, N+P-BURLOW and N+P-BURHIGH and CTR. The cyan colour indicates regions with very low denitrification rates compared to CTR. The orange colour indicates regions with very high denitrification rates compared to CTR.

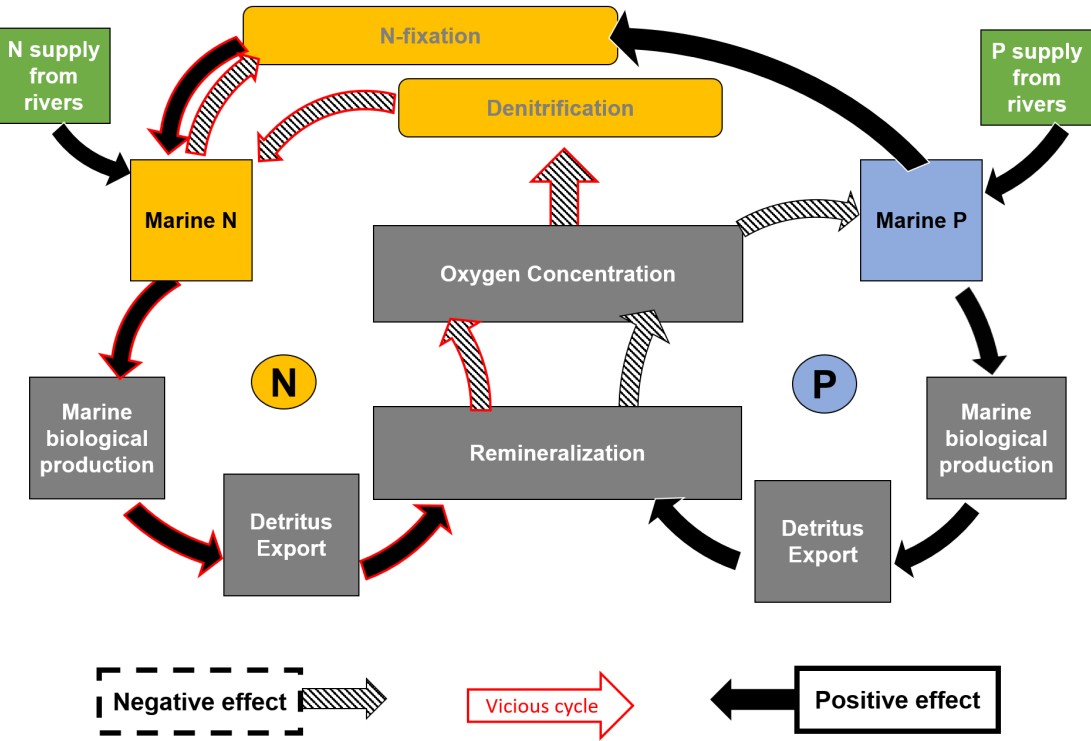

**Figure 10.** Feedbacks in the marine N and P cycle, due to riverine nutrient supply to the global oceans. Yellow colored boxes refer to processes concerning the marine N inventory. Blue refers to the marine P inventory. Green boxes show riverine nutrient input. The black arrows symbolise positive effects or feedbacks. The dashed arrows symbolise negative effects or feedbacks. Red bordered arrows symbolise processes involved in the vicious cycle (Landolfi et al., 2013).

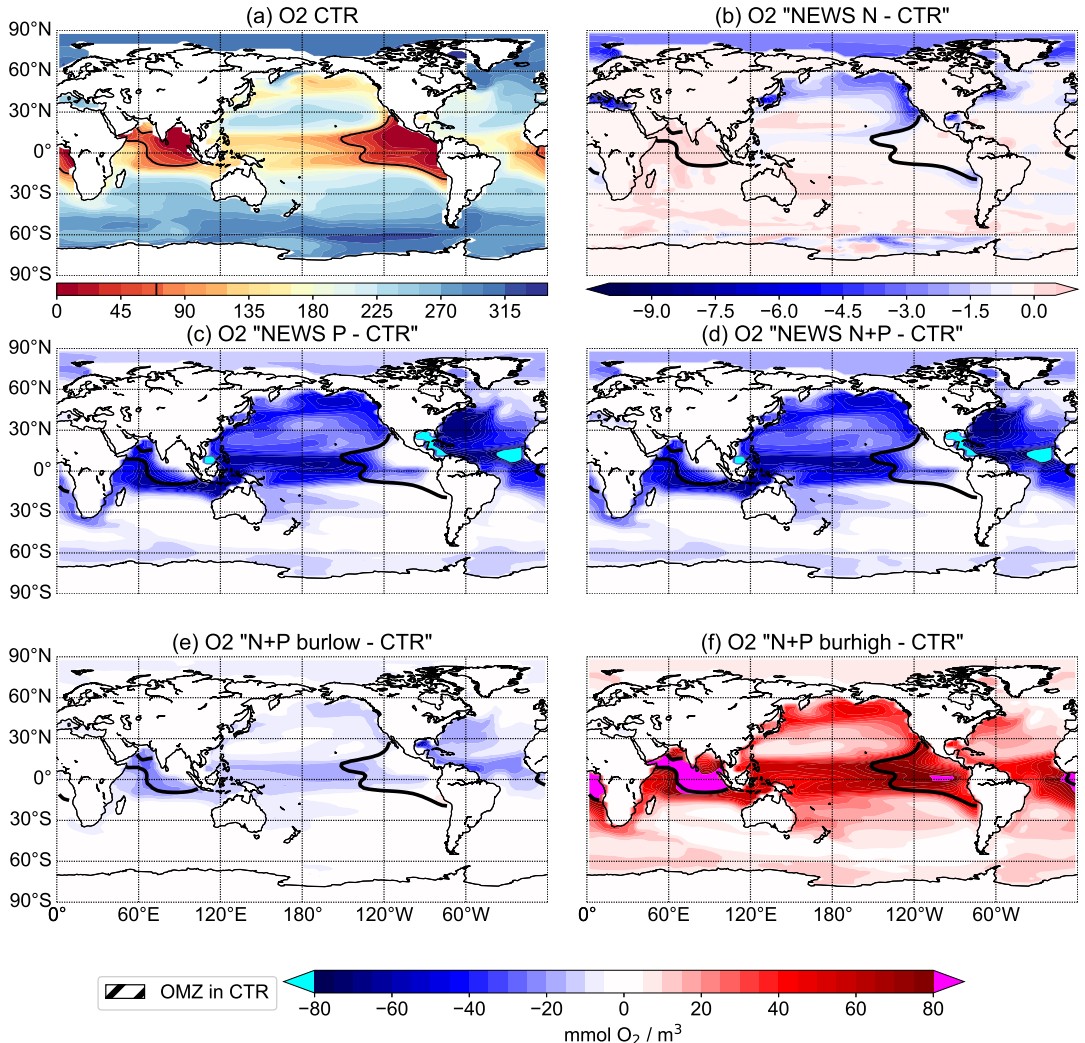

**Figure 11.** Oxygen concentrations at 300 m depth in mmol $O_2$ $m^{-3}$. (a) distribution of oxygen concentrations in the control simulation, difference in oxygen concentrations between the simulations NEWS (b), NEWS-P (c), NEWS-N+P (d), PP N+P-Burlow (e), PP N+P-Burhigh (f), respectively, and CTR. The black bold line shows the limits of the oxygen minimum zone at 302 m depth in the control simulation.

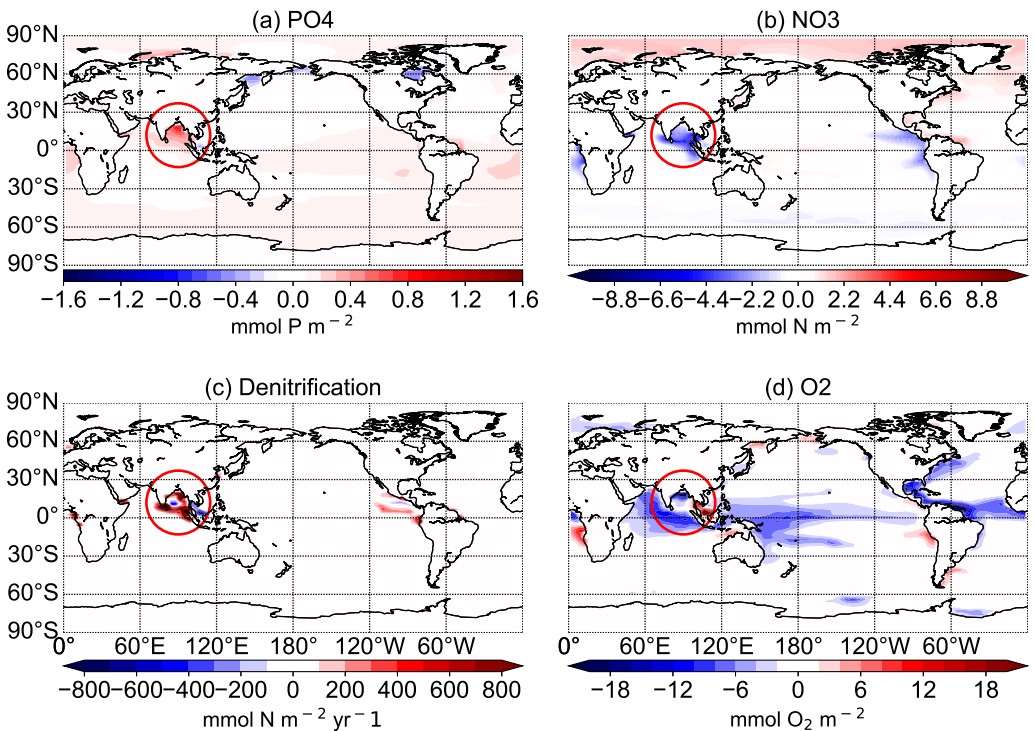

**Figure 12.** Difference between N+P-Burlow and CTR simulation for the average of the upper 300 m for PO$_4$ concentrations in mmol P m$^{-3}$(a), NO$_3$ concentrations in mmol N m$^{-3}$ (b) and O$_2$ concentrations in mmol O$_2$ m$^{-3}$ (d) and for the vertical integration of denitrification in mmol N m$^{-2}$ yr$^{-1}$ (c) at the end of the respective simulations over 10,000 years. The red circle shows the region of the Bay of Bengal.

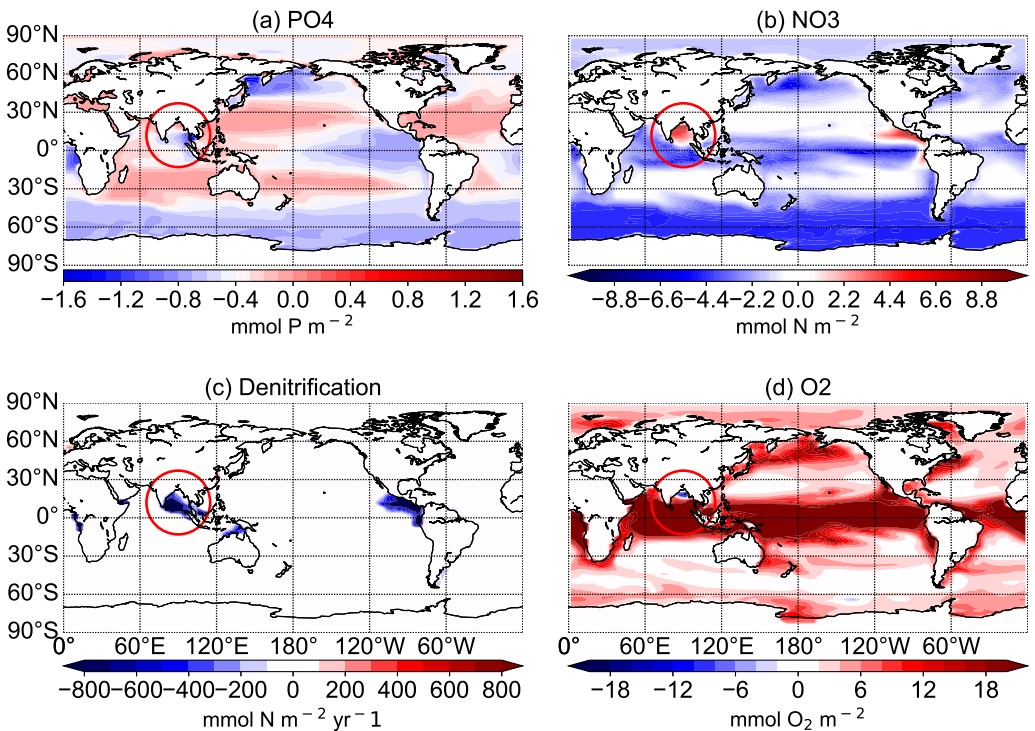

**Figure 13.** Difference between N+P-Burhigh and CTR simulation for the average of the upper 300 m for PO$_4$ concentrations in mmol P m$^{-3}$(a), NO$_3$ concentrations in mmol N m$^{-3}$ (b) and O$_2$ concentrations in mmol O$_2$ m$^{-3}$ (d) and for the vertical integration of denitrification in mmol N m$^{-2}$ yr$^{-1}$ (c) at the end of the respective simulations over 10,000 years. The red circle shows the region of the Bay of Bengal.

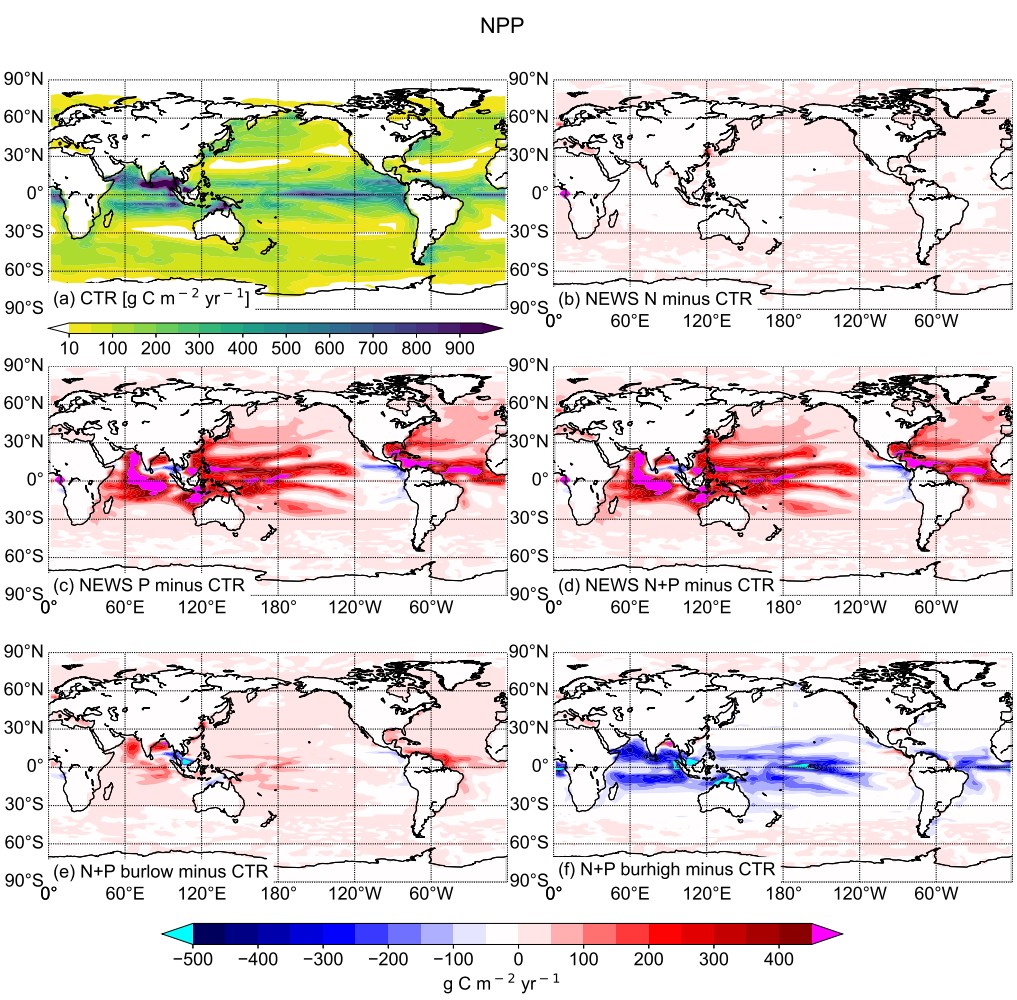

**Figure 14.** Vertically integrated production in g C m$^{-2}$ yr$^{-1}$.

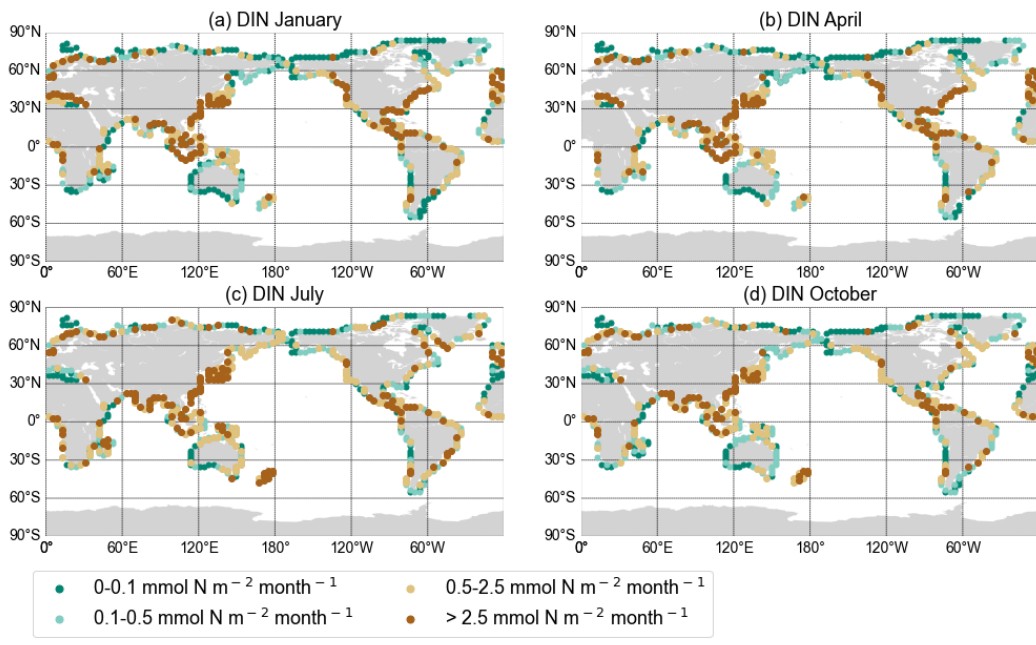

**Figure A1.** DIN export yield for each discharge point in mmol N m$^{-2}$ month$^{-1}$ from NEWS2 data set interpolated on the UVic grid for January (a), April (b), July (c) and October (d).

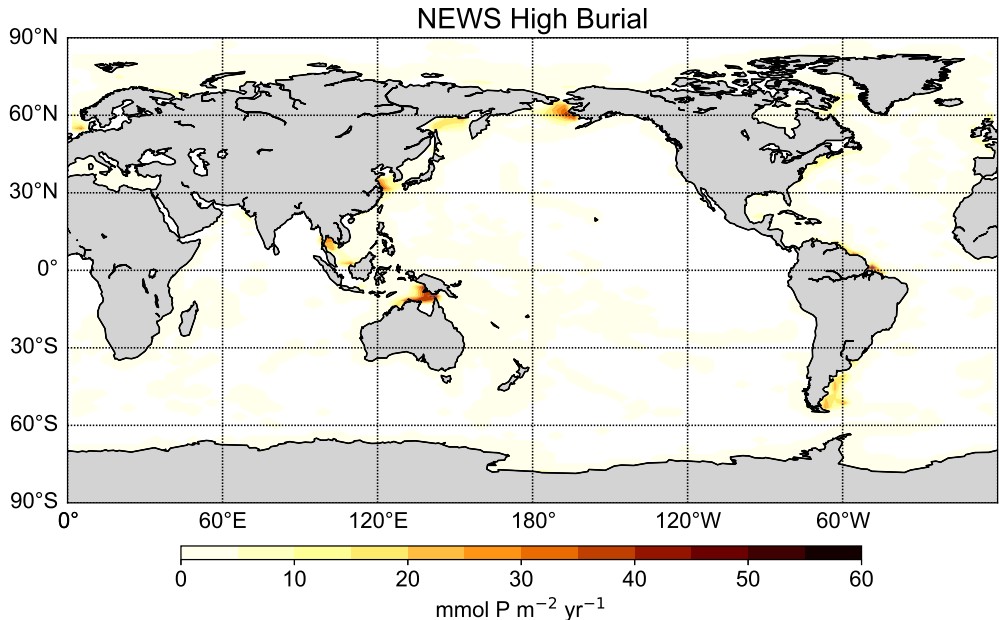

**Figure A2.** Burial flux of P in the N+P-BURHIGH simulation in mmol P yr$^{-1}$.

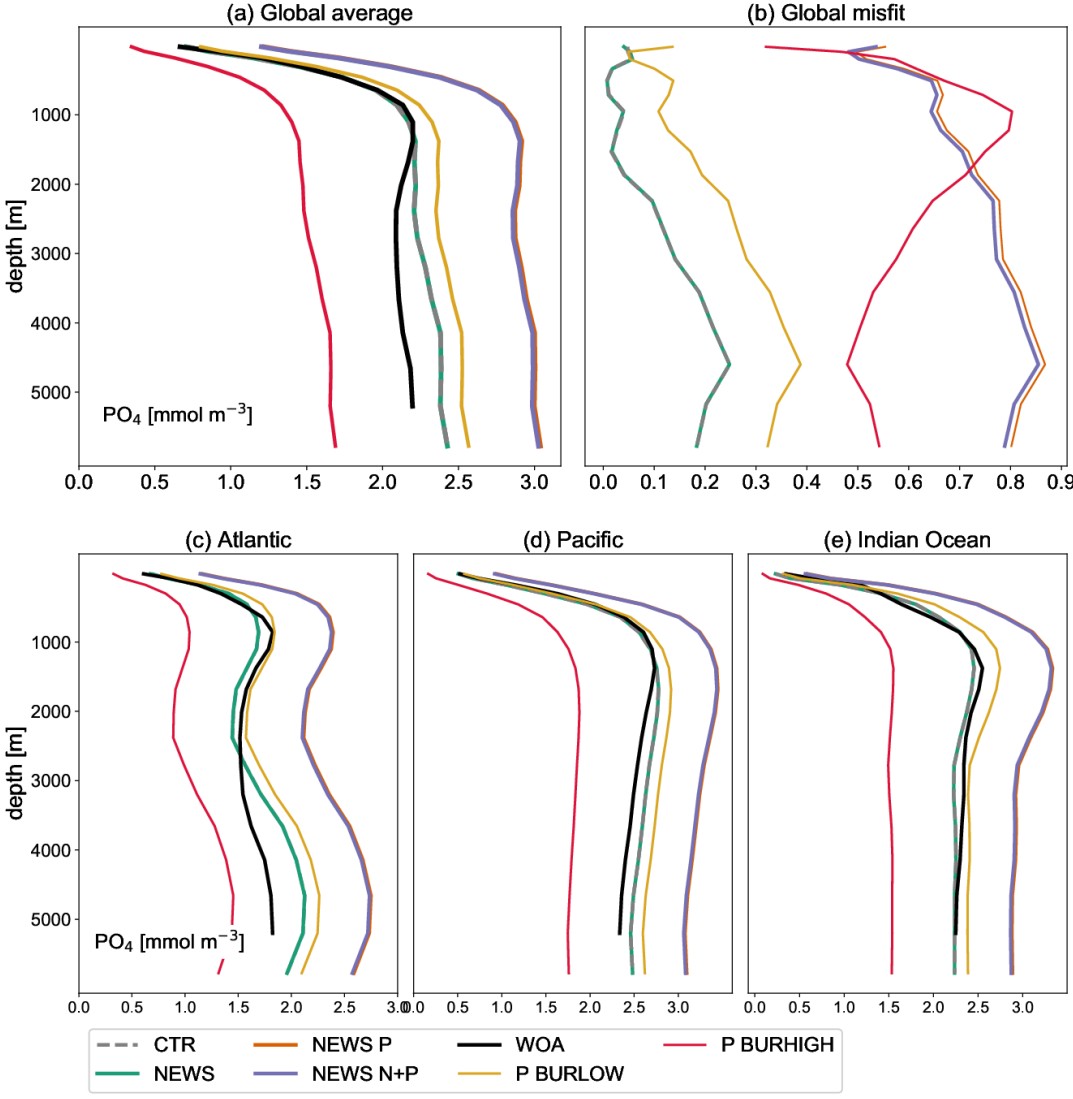

**Figure A3.** Global profile of $PO_4$ in mmol m$^{-3}$ in the simulations NEWS, CTR, NEWS-P, NEWS-N+P, NEWS-Burlow, News-Burhigh and from the World Ocean Atlas (WOA). (a) global average of $PO_4$ (b), Global profiles of misfit between each simulation and the WOA dataset (c), global average profiles of $PO_4$ in the Atlantic Ocean (d), global average profiles of $PO_4$ in the Pacific Ocean (e), global average profiles of $PO_4$ in the Indian Ocean (f).

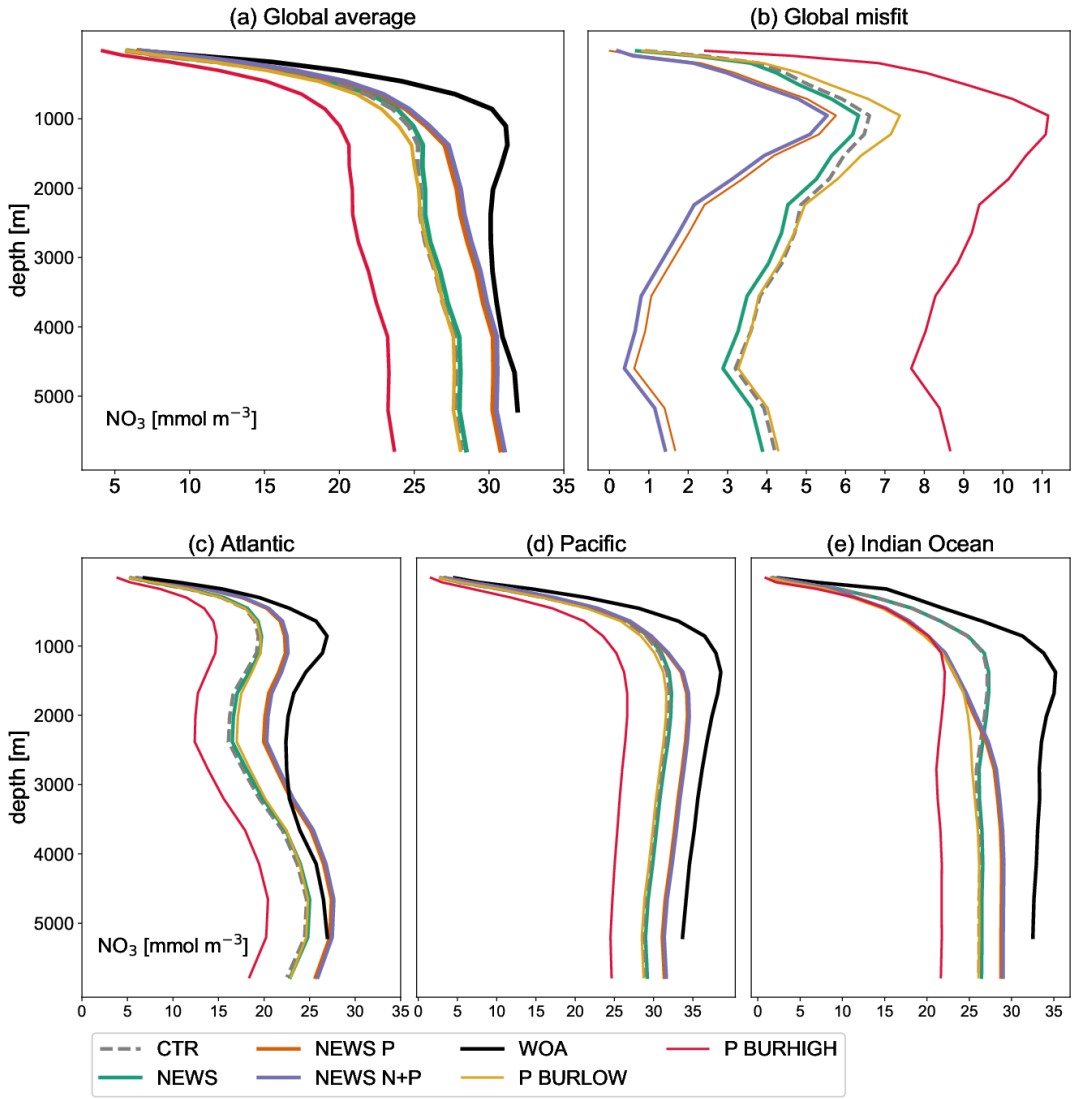

**Figure A4.** Global profile of $NO_3$ in mmol m$^{-3}$ in the simulations NEWS, CTR, NEWS-P, NEWS-N+P, NEWS-Burlow, News-Burhigh and from the World Ocean Atlas (WOA). (a) global average of $NO_3$ (b), Global profiles of misfit between each simulation and the WOA dataset (c), global average profiles of $NO_3$ in the Atlantic Ocean (d), global average profiles of $NO_3$ in the Pacific Ocean (e), global average profiles of $NO_3$ in the Indian Ocean (f).

*Author contributions.* MT developed the research concept in discussion with AO and DPK. DPK provided the initial model code, which was further developed, run, and analysed by MT. MT analysed the model output and visualised the results. MT wrote the manuscript with contributions from all co-authors.

*Competing interests.* The authors declare that they have no conflict of interest.

*Acknowledgements.* We gratefully acknowledge E. Mayorga, S. Seitzinger and their co-authors for making their database of Global Nutrient Export from WaterSheds 2 (NEWS2) available for our study. AO and DPK acknowledge funding from the European Union's Horizon 2020 Research and Innovation Program under grant 820989 (project COMFORT, "Our common future ocean in the Earth system — quantifying coupled cycles of carbon, oxygen, and nutrients for determining and achieving safe operating spaces with respect to tipping points") and OceanNETs (grant no. #869357). The work reflects only the author's view; the European Commission and their executive agency are not

responsible for any use that may be made of the information the work contains. This work was also supported by the German Research Foundation (DFG) as part of the research project SFB 754 "Climate-Biogeochemistry Interactions in the Tropical Ocean". We would also thank the GEOMAR's Biogeochemical Modelling group for many fruitful discussions.

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
