# Peer review of "Riverine nutrient impact on global ocean nitrogen cycle feedbacks and marine primary production in an Earth System Model"

_EGUsphere, 2024_

## Author Response (AR1)

**Dear referees,**

We would like to thank you for your very valuable and helpful comments and suggestions. We have taken them into account and modify our manuscript accordingly (see responses below). We hope, that we have addressed all your critics adequately.

**Comments from Referee 1**

*The relationship with Tyrrell (1999)*

**Referee1 (R1):** "Although the effect of river input of N is minor because of a negative feedback mechanism, the river input of P has a fundamental influence." Being critical, I would argue that the manuscript appears to just repeat this conclusion of Tyrrell (1999) after all. The authors should clarify (and discuss much more elaborately) what are common outcomes of those two studies, and what is the difference between the two. In the current manuscript, the advantage of using a 3D ocean model is not visible enough."

**Authors Response (AR):** Thank you for your critic. For our study, the work of Tyrell was starting point and framing. Tyrell started from Liebig's law of growth rate determined only by the availability of the most limiting substrate and defined then the 'proximate limiting nutrient' (PLN) representing the local limiting nutrient, and the 'ultimate limiting nutrient' (ULN) representing the nutrient whose supply rate forces total system productivity over long timescales.

It his study, Tyrell shows that the PLN and the ULN need not be the same: the ocean's PLN is reactive nitrogen while its ULN is simultaneously phosphate. Nitrate is the proximate limiting nutrient in surface waters; that is, the most limiting to instantaneous growth according to Liebig's law. Phosphorus, however, is predicted to be the ultimate limiting nutrient, whose rate of supply simultaneously regulates total ocean productivity.

So far, our results are indeed confirming the outcome of Tyrell. However, the model setup of Tyrell was a one-dimensional, two-box model of the global ocean, with the top layer representing the surface ocean down to the limit of the deepest wind-induced mixing during the year (the annual thermocline) and the bottom layer representing the deep ocean.

In our study, we used a global Earth system model (3D), not a box model, and hence also could analyze the global and regional distribution of nutrients. Our study explicitly focused on the nitrogen cycle and its internal feedbacks. We have started the experiment using only riverine nitrogen (see Tivig et al., 2021) and in the current

manuscript we describe the second phase of the experiment, where phosphorus was added via riverine export in addition to nitrogen.

In the revised manuscript we have explained more carefully the advantages and foci of our study compared to Tyrell (1999). (l. 117 ff )

**R1:**

*- The design of the experiments*
"The authors' general aim is not very clear to me. For example, which of the following three are the authors most interested in?

1. The effect of mean P concentrations. Most of the model results in this study is interpreted in the context of different mean P concentrations after all (i.e. contrast among the panels c,d <--> b,e <--> f of each global map figure).

2. The dynamics of an "open" system that has an explicit riverine input and sedimentary removal, in contrast to a closed system.

3. The biogeochemical dynamics in a 3D model (e.g. heterogeneity of riverine nutrient impacts), in contrast to a 1D model.

If 1, controlled experiments with different amounts of 'prescribed' total P inventory (with a closed system) would be better for more straightforward interpretation. If 2 or 3, however, experiments with 'forced' balance (i.e. the same amount as riverine input is "automatically" removed from the bottom grid cells of the model ocean) would be helpful to rule out the effect of different mean concentrations of P."

**AR:**

Thank you for this comment, which highlights, that we have not been clear enough about our research question.

After a first study with UVic, with riverine N input, where we studied the nitrogen cycle response in the coastal and open ocean on millennial timescales, we here complete the study by introducing riverine phosphorus in the same way. As the modelled ocean biogeochemistry of UVic does not comprise a full modelled phosphorus cycle (contrary to N), we at least had to construct a sink of P. The aim of this study was then, to study the nitrogen feedback reaction to the additional P. We have hopefully made this clearer in the revised manuscript. (l 109-113)

**RC:** *- The overall structure of the manuscript*

"The general structure of the manuscript seems to obscure its main purposes and messages. For example, the first item in the conclusion section is about P, and the first result section is also for P. Nevertheless, the manuscript has only one figure for the simulated P (Fig.3), while it has 7 figures for N."

**AR:** Thank you for your comment. See also the comment before:

Our research question was about N cycle feedbacks and its reaction to additional phosphorus from rivers. Therefore, the general distribution of P is interesting to understand changes in N cycle dynamics, leading to changes in N concentration. In the revised manuscript, have clarified our aims in the introduction and focused the conclusions on the main research questions (l. 106-108 and l. 440-457).

**RC:** - *Drift in the CTR experiment*
"L.192 says "the model was run for 10,000 years, starting from an already-spun-up steady state with the standard model version without riverine nutrients", and L.195 "a control simulation (CTR) was performed without riverine nutrients." If so, the N inventory ought to be quasi-constant throughout the experiment CTR. However, the result shows an obvious drift of the N inventory (Fig.4). What is the reason for that?"

**AR:** You are right, there is a shift in total N in the control simulation. We omitted to mention explicitly, that the spun-up model did not include benthic denitrification and the sub-grid scale bathymetry. We added these features in our CTR and NEWS simulations. Therefore, the model has to adapt to these new modules. We have added this to our model descriptions and add a sentence about the drift in N ( l. 215-217).

**R1:** - *Negative feedback via the RR-dependent burial rate?*

"In the N+P-BURHIGH experiment, the resultant lower concentration of $PO_4$ and $NO_3$ at the surface would lead to lower RRp (P rain rate) to the sediment, which would result in smaller BURp eventually. If so, this process will bring a negative feedback to stabilize the P inventory because a constant river input of P is given to the ocean. Therefore, I speculate that the P inventory should follow an asymptotic evolution that approaches a new steady state. Nevertheless, the N inventory continues to decrease very linearly even after the 10k years (Fig.4). What is the reason for that? N decreases due to another mechanism, although P is (or is becoming) stabilized (I cannot judge this because there is no time-series plots for the global P inventory)? Or, the 10kyr model integration is far too short to obtain a new steady state?"

**AR:** The total P burial flux (BURp) is decreasing in the high burial scenario with time, but is still much higher than the low burial rate even at the end of the 10000 years simulation. No new steady state has been reached and P and N are still decreasing at the end of the 10kyr. In order to reach a steady state here, the simulation should have been much longer indeed than we should run UVic for. However, the trends can be seen in the timeseries, that we have also included in the revised document (Figure 3). We have elaborated on this question in our revised manuscript. Unfortunately, we do not have time to rerun the whole simulation to prove it (l. 284-287).

What can nevertheless be seen in the Figures here showing time-series for the global P inventory and P fluxes, is the increase in P flux in the high burial simulation

**R1:** - *Burial of N*
"Although the burial flux of N would be too large to be regarded as negligible (e.g. Tyrrell, 1999; Gruber & Galloway, 2008; Voss et al. 2013), the factor is not even touched in the manuscript. Why did not the authors employ a N-burial scheme that is similar to the method for P-burial? At least some quantitative discussions should be included to evaluate the influence of N burial on the authors' arguments."

**AR:** Sorry that this was not clear. As in previous studies with UVic (e.g. Somes et al., 2017; Tivig et al., 2021), we simulated sedimentary N-loss according to an empirical transfer function based on organic carbon sinking flux to the sediments and bottom-water dissolved oxygen and nitrate. We used the equations from Bohlen et al. (2012). Therefore, there is indeed a N-burial scheme in our model setup. We have been more explicit on this in the description of the model setup.

**R1:** - *Model resolution*
"It is expected that the (horizontal) spatial resolution of the model (1.8 x 3.6 degrees) is too coarse to represent small-scale processes in coastal oceans. In particular, in the context of the manuscript, the influence of the resolution on the model representation of the communication between coastal oceans and open oceans would be an inevitable issue. Discussion on the robustness or uncertainty of the model results should be added from this viewpoint."

**AR:** We have added a paragraph on limitations and uncertainties of our model setup (see also RC2). This new text includes e.g. the following points:

- Sedimentary N-loss is simulated according to an empirical transfer function based on organic carbon sinking flux to the sediments and bottom-water dissolved oxygen and nitrate. We apply the empirical function from Bohlen et al. (2012). Since our coarse resolution model does not fully resolve narrow continental shelves and coastal dynamics, it underestimates sedimentary N-loss in these regions (see also Somes et al. 2017).

  Since continental shelves are not well resolved in the model, a subgrid-scale bathymetry parameterization is included (Somes and Oschlies, 2015).

- Still there are processes, that are not resolved: e.g. nutrient retention on the shelf / in coastal sediments; buffer effects of the coasts that could be parametrized like in Sharples et al (2017) and Izett and Fennel (2018). No groundwater water discharge to the oceans;

- the current study does not incorporate all the feedbacks that may potentially impact the N cycle (e.g., stoichiometric variability of phytoplankton, anthropogenically induced changes in ocean acidification and iron deposition)

================================================

**[Other major issues]**

*\* l.183*
"Some brief explanations (e.g. algorithm) as to how \*BURc\* is calculated would be required."

**AR:** We have added text (L.203ff) describing how BURc is computed from the modelled detritus export in terms of carbon (following Kemena et al. (2019):

On the shelf and continental margin:

BURc = 0.14 RRc^1.11

In the deep sea:

BURc = 0.014 RRc^1.05

RRc is a variable from UVic

*\* Section 2.3*
"Some plots showing the global distribution of riverine nutrient fluxes will be surely appreciated."

**AR:** Sorry, we omitted this here. We have now added them analogous to our publication with riverine N only (Tivig et al., 2021) for riverine P (Figure 2).

*\* l.243, .... N+P-BURLOW appears most similar to observed present-day oceanic conditions.*
"The authors would need to demonstrate an explicit model-data comparison to support this, if they keep this sentence in the next version."

**AR:** Thank you for this comment. Our statement is based on comparisons with the data from the World Ocean Atlas (WOA). The depth profiles (see figure below) show, that the scenario with low P burial rates is closest to the WOA in terms of global profiles of P, at least for all simulations including riverine P input. Misfit is shown on the top right panels. As UVic is globally underestimating N concentrations especially in the upper 2000 m of the ocean (compare also Keller al., 2012), for the global N profiles, the two simulations including P from rivers but without a burial function

show a better correspondence to WAO profiles. But these two simulations are much more unrealistic in terms of P profile. We include these figures in the manuscript as supplemental material. (compare also l. 270-272)

*\* Figure 5*
"It is difficult to see whether the black "global sum of fluxes" lines are consistent with Fig.4 or not. Probably, a thin dotted line or similar to show the zero level (of the right-hand y axis) will be appreciated."

**AR:** We have added the zero-line in the current version (Figure 5).

*\* Figure 5c*
"A blue dashed line for "DIN from rivers" seems to be missing."

**AR:** Thank you very much for this comment. The dotted line for the riverine N input is missing in the bottom panel. **AR:** We have added the line in the current version (Figure 5).

*\* Figure 6, the color bar*
"The color scheme should be symmetrical around zero (i.e. should be white around the zero for both sides)."

**AR:** We have rearranged the figures with a new colorbar (Figure 6 and 7).

*\* l.290, iron availability also plays an important role.*
"More description of the role of iron in the "vicious cycle" context would be appreciated."

**AR:** Thank you for you remark. Iron limitation is a limiting factor for the growth of diazotrophs and hence N2-fixation (Landolfi et al. 2013; Moore et al. 2009). We have added more descriptions of the mechanisms of potential decoupling of denitrification (N loss) and N2-fixation (N gain) and its consequences on the global fixed nitrogen inventory (l. 327 ff).

*\* the last paragraph of section 3.3, especially l.322-332.*
"I would say it is very troublesome to follow the discussion in this part, because the readers are forced to 'jump' about from one figure to another (i.e. Fig.10 --> 7 --> 12). Besides, it is difficult to follow the discussion with small global plots because sometimes it is not obvious which region is the target of discussion. Therefore, for better illustration, I would suggest that the authors make a new figure that specializes in the particular discussion."

**AR:** We have made two new figures (Figure 13 and 14) to illustrate our argumentation and restructure this paragraph, in order to make it easier to follow the discussion.

*\* Figure 12*
"This elaborate feedback diagram is referred to in the main text only once, and in a not-really-visible way, although it seems to have great potential to promote communities' understanding. I would recommend that the authors should add a substantial amount of description, explanation and discussion to make the best use of this diagram. Otherwise, the diagram itself is not very intuitive, and perhaps it is not necessarily needed in this particular manuscript."

**AR:** As the feedbacks in the N cycle are the main focus of this study, we have developped this part of the manuscript differently. The feedback diagram is introduced earlier in the manuscript and better integrated in the text (e.g. l. 322, 385ff).

*\* Figure 12 (cont.)*
"If the authors keep this diagram in the next version, I would suggest that each item in the diagram should not have variation or tendency, namely "Oxygen concentration" instead of "Oxygen reduction", for example. Another example (in a different context) is "temperature" -->(-) "ice cover" -->(+) "albedo" -->(-) "temperature", where one can see this represents a positive feedback loop because there are two anti- correlations. Otherwise, such a diagram can be confusing or misleading. Indeed, the current Fig.12 seems to have at least one error. With "Marine N" -->(-) "Decrease in N fix", I guess that the authors should have intended to indicate "Marine N" -->(-) "N fix"."

**AR:** Thank you for these critics and suggestions, we have taken them into account in the revised manuscript and corrected the figure (Figure 11).

*\* l.350*
"I would doubt the significance of this item (the 1st bullet point). It would be obvious that the accumulation speed of P decreases with the burial rate. If the authors find another implication from those numbers, it should be added or described more clearly. For example, if the simulated accumulation rates themselves have any significance, the authors should discuss the values, possibly by comparing with independent numbers from another literature."

**AR:** Thank you for this comment. Indeed, the first bullet point is not exactly the answer to the first question we rise in the introduction ("How does riverine N and P input, rather than riverine N input alone, affect the representation of ocean biogeochemistry including marine primary production in the model?") and the numbers given here are not really relevant for this question. Therefore the bullet point has been deleted.

*\* l.362, This can be attributed to the so called "vicious cycle", triggered here by higher N and P supply in proximity to low oxygen regions.*
"I would doubt this hypothesis for the following two reasons.

- For the vicious cycle, N2-fixation and denitrification need to be spatially coupled. However, in the model results, the spatial distribution of the N2-fixation anomaly and that for the denitrification do not overlap with each other (Figs.9,10)."

**AR:** You are right, our answer here is much too short and not completely right. There is not much overlap between the regions of high denitrification and N2-fixation rates in our model. Therefore the "classical" vicious cycle induced by these to processes is not that relevant. But in the simulations, where enough P is added from rivers with low or no burial, in regions where denitrification is active (Figure 10 c,d,e), we can find a net N loss, with denitrification fueled by increased O2 loss due to primary production increase, which is triggered by additional P (e.g. combined Figure 13). Independently of the N input, denitrification is higher than in the simulations without riverine P input. The paragraph has been partly rewritten (l. 449 ff).

**R1:** In NEWS_burhigh, where denitrification is much lower due to higher oxygen concentrations (again due to decrease in NPP rates), the N loss is probably a consequence of the decreased N2-fixation rates, as the main source of bioavailable N in the global ocean.

- "The vicious cycle represents a runaway loss of fixed N by stimulated N2-fixation and denitrification (Landolfi et al., 2013). Therefore, for example in the NEWS_N+P experiment that has positive anomalies of N2-fix and denitrification (Fig.9d,10d), the total amount of N should have decreased due to the runaway loss if the vicious cycle had been the main mechanism. However, actually the total N inventory increased in NEWS_N+P, which appears to be against the hypothesis. Similarly, in the N+P-BURHIGH experiment, the positive feedback would have brought the increase of N inventory (and that happened indeed only in the Gulf of Bengal and the eastern tropical Pacific; Fig.7f), but the overall change in N inventory was again opposite."

"Therefore, I would suspect that the overall increase or decrease in the N inventory needs to be explained by another mechanism. Clarification by the authors will be highly appreciated."

**AR:** Thank you for this comment. N-fixation increases due to the increase in P, but in the simulation with high burial rates, P and N decrease globally, as well as denitrification. Here, not the vicious cycle is responsible for the N loss, but the limitation of N-fixation.

The vicious cycle can be observed locally for example in the Bay of Bengal (Figure 13 and 14). We will rephrase our conclusions to make the above points clearer.

*\* l.369, While this result is linked to our model configuration, it is nevertheless relevant for the real ocean.*

"More specific descriptions about this sentence should be given. For example, what is expected model limitations? What is still valid and robust in terms of the connection to the real ocean?

**AR:** The underlying UVic simulation has been evaluated before (Keller et al. 2012) including its limitation compared to previous simulations and observations. Additionally, we added a new module for riverine nutrients, which has its own limitations, using nutrient data with limitations (NEWS2, Mayorga et al., 2010). As discussed in Tivig et al. (2021), our module is also limited by the model resolution, not able to adequately resolve coastal and shelf processes, like nutrient retentions for example." In the new section on limitations, we have included more discussions on the robustness of our findings with regard to the real ocean.

==============================================

**[Minor points]**
*\* l.54, citepTyrrell99*
"A potential typo. " This has been corrected in the revised manuscript.

*\* l.88, ration*
"A potential typo."  This has been corrected in the revised manuscript.

*\* l.126, ... they are not limited by NO3 nor by a maximum NO3 concentration...*
"This description does not make sense to me. Additional explanation would be appreciated."

This has been corrected in the revised manuscript.

*\* l.156, Following other studies, we decided to include dissolved organic and inorganic P (DOP, DIP), as well as 45 % of total particulate P (TPP)*

"What is the reason for the 45%? What studies did you follow?"

**AR**: Colman and Holland (2000) and Ruttenberg (2003) found that approximately 25-45 % of total particulate phosphate is reactive. Therefore, we used the 45 % of NEWS2 PP and included them to the riverine P flux. We have stated this more clearly in the revised manuscript (l. 167 ff).

*\* Table 1*
"Only(?) in this table, the unit for the fluxes is 'g', although 'mol' is used in the other parts of the manuscript."

**AR**: Thank you very much for this indication! We harmonized the units in the revised manuscript.

*l.203, (Table 1*

"A potential typo. " This has been corrected in the revised manuscript.

*l.292, N2-fixation is influenced by P only.*

"The authors' intention of this sentence is not clear to me."

AR: Thank you for the comment, our sentence is not clear here. We meant, that iron limitation for N2-fixation stays constant in our simulation (iron is kept fix), so only the P inventory, has an influence on changing N2-fixation rates. The sentence has been rewritten.

*Fig.11, 302 m*

"A potential typo. 300 m?"   This has been corrected in the revised manuscript.

*l.361, by -1.5 % in N+P-BURLOW and -18.2 % in N+P-BURHIGH*

"Are these numbers consistent with Fig.4? If I understand correctly, Fig.4 would indicate "-15% in N+P- BURLOW and -30% in N+P-BURHIGH."

AR: Yes, thank you again. The numbers are correct, but they indicate the difference in the global N inventory compared to Control and not compared to the start of the simulations. The change in the inventory compared to the start of the simulations is then:

| CTR | -17,4 % |
|---|---|
| NEWS_N | -16,1 % |
| NEWS_P | -8,5 % |
| NEWS_N+P | -7,9 % |
| N+P_Burlow | -18,8 % |
| N+P_Burhigh | -32,4 % |

The sentence in the manuscript has been changed.

[Figure]
Reply
**Citation**: https://doi.org/10.5194/egusphere-2024-258-AC1

**Comments from Referee 2**

**Major issues:**

*Referee Comment (RC):* **"***The authors have acknowledged that including burial functions in the model is essential to balance the nutrient inputs. However, the model's burial sink is only applied to P and not N. The authors should provide a clear explanation for this decision. Nevertheless, I recommend that they simulate the model by including N burial.***"**

**Authors Response (AR):** Sorry that this was not clear. Like in previous studies with UVic (e.g. Somes et al., 2017; Tivig et al., 2021), we simulated sedimentary N-loss according to an empirical transfer function based on organic carbon sinking flux to the sediments and bottom-water dissolved oxygen and nitrate. We used the equations from Bohlen et al. (2012). Therefore, there is indeed a N-burial scheme in our model setup. We have been more explicit on this, in the description of the model setup (l. 182).

*RC -*" *I appreciate the three different burial scenarios presented in the simulations, which include no, low, and high burial. However, I feel that the study's objectives, as defined in the introduction section, do not necessarily require all three experiments. Additionally, as with any geoscientific model, the goal should be to replicate the real-world system's behavior to generate insights into the problem being studied. While the authors noted that one of the simulations is closer to observation, there is no evaluation of model data misfit. To gain better insight into burial parameters, I suggest analyzing the simulations with present-day observations. This would allow us to determine which scenario best simulates the spatial dynamics of P and primary production.*"

**AR:** As we want to analyze the effect of P on nitrogen cycle and N feedbacks, it seems appropriate to use the 3 experiments: one without burial, one with low ("near realistic"?) burial and one with high burial. The three scenarios lead to different P concentrations and by this we can also see, how these different P concentrations influence the N cycle.

Thank you, however, for pointing out a lack of comparison to real world results. We have evaluated our model output with the data of the World Ocean Atlas (WOA) at least for the depth profiles (see figures A3 and A4 included in the new manuscript as supplemental material). The depth profiles show, that the scenario with low P burial rates is closest to the WOA in terms of global profiles of P, at least for all simulations including riverine P input. Misfit is shown on the top right panels. As UVic is globally underestimating N concentrations especially in the upper 2000 m of the ocean (compare also Keller al., 2012), for the global N profiles, the two simulations including P from rivers but without a burial function show a better correspondence to WAO profiles. But these two simulations are much more unrealistic in terms of P profile.

*RC:* - *"The UVic model is a useful tool for simulating global biogeochemical cycles. However, the manuscript recognizes that the model has certain limitations, such as its coarse resolution and simplified assumptions. The authors should discuss how these limitations might affect the simulated impacts of riverine nutrient inputs on marine ecosystems."*

**AR:** Thank you again for your very relevant comment. In the revised manuscript we refer to previous studies and discuss more thoroughly limitations of the model and the experimental set-up (**Section 4**).

- *"The study relies on the Global Nutrient Export from WaterSheds 2 (NEWS 2) model data for riverine nutrient inputs. The authors note the large uncertainties in real and modeled nutrient fluxes, which could influence the study's conclusions. I wish to see some more discussions in this direction."*

**AR:** Thank you for this comment. In the revised manuscript we elaborate more on the uncertainties resulting from the use of the NEWS2 model data set of riverine nutrients. We refer to Mayorga et al. (2010), Dumont et al. (2005) and others, who evaluated the individual models for the river export. They generally found, that the global export estimates were similar to previous publications but also reported some limitations. We also mention, that NEWS2 includes anthropogenic nutrient sources, while our UVic model is based on preindustrial conditions for example for $CO_2$. Finally, we have to consider the coarse resolution of coastal regions in UVic, with no explicit simulation of the coastal processes. Therefore, nutrient retention on the coastal shelf is not included and eventually riverine nutrient export to the open ocean is overestimated in our experiments.

*RC:* - *"The study's focus on millennial-scale simulations provides essential insights into long-term trends, but it may overlook shorter-term variability and responses of marine ecosystems to nutrient inputs. While the manuscript acknowledges the influence of human actions on riverine nutrient flows, discussions on shorter-term variability and human influences on riverine nutrient exports and marine ecosystems. Including some discussions about the impacts of climate change over the century scale could offer a more nuanced view of human influences on marine ecosystems."*

**AR:** While the riverine nutrient export from NEWS2 includes the anthropogenic component, based on the year 2000, we did not focus on the question in our experiment and the millennial-scale simulations do not provide much insight into the short time variability. Nevertheless, the question of the influence of human activities on the marine N cycle is the starting point for our studies. The background for these activities was the question about the influence of anthropogenic activities on the coastal oceans, the rivers being one component of this question. Therefore, we thank you for your remark and have included a remark on human influence on ocean biogeochemistry.

**RC:** - *"I like the schematic in Figure 12. However, I need to include a discussion on the feedback mechanism. By integrating riverine phosphorus inputs and examining their impact on nitrogen cycle feedback, this manuscript provides new insights into the interplay between nitrogen and phosphorus cycles in the ocean - how phosphorus availability can alter nitrogen fixation and denitrification rates, offering a more detailed understanding of the feedback mechanisms that regulate global biogeochemical cycles. Therefore, the schematic is worth some attention."*

**AR:** Thank you for this comment, which complements a similar comment of Referee 1. In the revised manuscript, Figure 11 (before 12) is put more into focus and discussed in more detail.

**Other comments:**

- *Global maps of nutrient inputs would help understand the spatial dynamics of their influence.*

**AR:** Yes, we have included maps with riverine nutrient input.

- *In table(s), figures, and conclusion, the authors use 'g' in the unit, but in the result section, they use 'mol.' It would be nice to have consistency throughout the text.*

**AR:** This has been harmonized in the revised manuscript.

- *Figure 3: PO4 concentration in all panels, rather than the difference in concentration, would be more useful to understand the spatial dynamics.*

**AR:** We have changed this figure in order to show absolute concentration for PO4 (Figure 4).

- *Figure 4: I suggest also including a time series of P in another panel.*

**AR:** Yes, the timeseries is included in the revised manuscript (Figure 3).

- *L345: In this follow-up study to Tivig et al. (2021), a new component was added to ... A sentence here about what Tivig et al. (2021) did would be useful.*

**AR:** Yes, we have included more explanations in the revised manuscript (l.431 ff).

**Minor issues:**

- *L5: 'additionally include' sounds redundant.* **AR:** Yes, has been corrected in the revised manuscript (l. 5).

*- L88: In a previous study, ... the citation is missing.* **AR:** Yes, has been corrected in the revised manuscript (l.55).

*- L27-28: very long sentence, hard to read.* **AR:** This has been corrected in the revised manuscript (l. 26-27).

*- L127: Not clear what is said.* **AR:** Sorry, this has been clarified in the revised manuscript (l. 138).

*- L156: "Following other studies ..." Which other studies? The ones mentioned above?* **AR:** Sorry for leaving this out, we have named the studies more explicitely in the revised manuscript (l. 168).

*- L157: please specify the study(ies) to determine 45% TPP. Did you try to tune it?*

**AR:** Colman and Holland (2000) and Ruttenberg (2003) found that approximately 25-45 % of total particulate phosphate is reactive. Therefore, we used the 45 % of NEWS2 PP and included them to the riverine P flux. We have explained this more thoroughly in the revised manuscript.

No, we have not exactly tried to tune it, but did some experiments with the NEWS2 dataset before launching the model and found that using DP + 45% of PP gave us total reactive P amounts for the river export closer to the ones we found in the literature as cited in the manuscript. (l. 170)

*- L204: '... previous studies ...' citations are missing.* **AR:** Citations have been included in the revised manuscript (l. 228).

*- L234-235: Has the spatial variance changed substantially from the initial conditions?*

**AR:** We here calculated the standard deviation of the concentration of phosphate in the model and yes, it was somewhat higher at the beginning of the simulations (for NEWS_N 0.650 to 0.646 mmol P m$^{-3}$), but as this reduction is parallel in CTR and NEWS (see figure below), we assume that it is due to the adaptation to the new features (benthic denitrification and subgrid bathymetry).

[Figure]

*- L285: You mean 'the Bay of Bengal'?*  **AR:** Yes, thank you. This mistake has been corrected here and at other places.

*- L292: 'N2-fixation is influenced by P only.' And?*
**AR:**  We meant, that iron limitation for N2-fixation stays constant in our simulation (iron is kept fix), so only the changing P inventory, has an influence on changing N2-fixation rates. Therefore, changes in N2-fixation can be attributed to P. This will be clarified in the revised manuscript (l. 330).

*- Figure 11: (& Figure 7) 300 and 302m-which one is correct?*
**AR:** We have corrected it to 300m in the manuscript for Figure 11. The former Figure 7 has been changed to the average of the upper ocean levels (Figure 6).

*- Figure 13: 'Vertigal' typo?*  **AR:** Yes, thank you. The typo has been corrected (Figure 14).

---

## Author Response (AR2)

Response to referee comments (Minor Revisions)

**Suggestions for revision or reasons for rejection**

The manuscript acknowledges the limitations of the UVic model, specifically its coarse resolution and the lack of consideration for coastal processes. Adding a brief discussion, perhaps a couple of sentences, on how these limitations might impact the results would enhance the rigor of the manuscript.

Thank you very much for this comment. We now have expanded the section "Discussion and limitations" and added more details on the limitations of our model and how they might impact the results of our study (Line 400-412).

Additionally, it would be beneficial to discuss the potential impact of other feedback mechanisms not included (phytoplankton stoichiometry, changes in ocean acidification, and iron deposition) in the current model.

Thank you for this remark. Indeed, in the last version we just mentioned other feedbacks, without explaining their potential impact. We have now elaborated more on the possible influence of phytoplankton stoichiometry and iron deposition. The role of ocean acidification, although very relevant for the modern ocean's biogeochemistry, is not directly linked to our current research question. Therefore, we finally decided to leave it out (Line 415-425).

Please review the manuscript comprehensively to correct any minor grammatical errors and enhance overall clarity and readability. Here are a couple of examples from the abstract.
L3: Internal feedbacks regulate ...
Feedback of what?
L8 ... addition of bio-available phosphorus alone or together with nitrogen affects ...
Addition to what?
* * *
We now have reviewed the manuscript and corrected grammatical errors and typos, like those you mentioned for the abstract.